# A modeling framework for determining modulation of neural-level tuning from non-invasive human fMRI data

Patrick Sadil [1✉], Rosemary A. Cowell[1,2] & David E. Huber[1,2]

Many neuroscience theories assume that tuning modulation of individual neurons underlies changes in human cognition. However, non-invasive fMRI lacks sufficient resolution to visualize this modulation. To address this limitation, we developed an analysis framework called Inferring Neural Tuning Modulation (INTM) for "peering inside" voxels. Precise specification of neural tuning from the BOLD signal is not possible. Instead, INTM compares theoretical alternatives for the form of neural tuning modulation that might underlie changes in BOLD across experimental conditions. The most likely form is identified via formal model comparison, with assumed parametric Normal tuning functions, followed by a non-parametric check of conclusions. We validated the framework by successfully identifying a well-established form of modulation: visual contrast-induced multiplicative gain for orientation tuned neurons. INTM can be applied to any experimental paradigm testing several points along a continuous feature dimension (e.g., direction of motion, isoluminant hue) across two conditions (e.g., with/without attention, before/after learning).

[1] University of Massachusetts, Amherst, Amherst, USA. [2] These authors jointly supervised this work: Rosemary A. Cowell, David E. Huber. ✉email: psadil@umass.edu

Neurons in mammalian sensory cortex often exhibit tuning functions, responding selectively to a narrow range of values of a stimulus feature such as line orientation[1]. Moreover, experimental manipulations of perceptual or cognitive state can modulate these tuning functions, providing crucial insights into the neural mechanisms of perception and cognition. For example, spatial attention achieves performance gains by multiplicative rescaling of orientation tuning functions in macaque V4[2], rather than through a bias shift (additive shift) or increased selectivity (sharpening). As another example, Fig. 1a shows a neuron in mouse primary visual cortex[3], tuned to motion direction, that appears to undergo multiplicative rescaling as spatial frequency is manipulated (Fig. 1a). However, single-cell electrophysiology is rarely feasible in humans, and non-invasive techniques such as functional magnetic resonance imaging (fMRI) reflect the activity of many cells: a patch of striate cortex the size of a typical fMRI voxel (2 mm³) reflects the activity of approximately 300,000–500,000 neurons[4]. Nonetheless, fMRI studies reveal feature-selective tuning in voxels (e.g., in which a voxel's BOLD response varies systematically as a function of a well-defined feature such as stimulus orientation)[5,6], which is assumed to arise from a non-uniform distribution of tuning preferences across the neurons contributing to a voxel[7].

Like neural tuning functions, voxel tuning functions are modulated by manipulations of perceptual state[8–11], and so it is tempting to infer the form of *neural-level* tuning modulation directly from observed changes in voxel tuning. But, as many have acknowledged, directly inferring neural behavior from voxel behavior might be misleading[10,12–14]. The relationship between voxel and neural tuning functions presents a many-to-one inverse problem: the same voxel tuning function can arise from many different combinations of the shape of the underlying neural tuning functions and the distribution of neurons across different preferred stimulus values (e.g., the number of neurons preferring vertical, horizontal, or oblique orientations). Thus, fMRI tuning does not uniquely specify tuning at the neural level[14].

Rather than *quantitatively* solving this inverse problem, which is probably not possible, we developed a framework that allows drawing *qualitative* conclusions about changes in neural level tuning from the Blood Oxygen Level Dependent (BOLD) signal. We term the framework Inferring Neural Tuning Modulation (INTM). Building on encoding models[15–17], INTM assumes that voxel tuning emerges from the combined activity of feature-tuned neurons. Unlike many existing encoding models, which assume that voxel activity is driven by a discrete number of neural "subpopulations", INTM conceptualizes the underlying neural preferences as existing along a continuum (Fig. 1b). This continuous distribution resembles population receptive field mapping, but whereas population receptive field mapping is generally concerned with uncovering the quantitative parameters of tuning (e.g., the size of receptive neural fields, or the population's preferred location in the visual field)[18], INTM is designed to uncover the modulation of tuning. To do this, INTM leverages the observation that some forms of modulation produce patterns of BOLD data that are distinct from other forms (Fig. 1c, d), and so the BOLD data provide information about which form of tuning modulation is most likely. This framework encompasses many kinds of analyses, but, in short, INTM prescribes building multiple encoding models of voxel data that are parameterized by continuous distributions at the neural level, followed by comparison between the models in light of the observed BOLD data to uncover the most likely form of tuning modulation.

We validated INTM by applying it to a new dataset for which the "ground truth" modulation of tuning was known from single-cell recordings: changes in visual contrast induce multiplicative scaling in orientation-tuned neurons[19–21]. This simple test-case

demonstrates the advantages of INTM, considering that existing methods, if interpreted as indicating the underlying neural tuning functions, can easily reach the wrong conclusion about the form of neural tuning modulation when applied to changes in visual contrast c.f.,[14,22]. We used INTM to design two complementary analyses that differ in the assumptions required for the statistical models underlying the analyses. The first analysis uses Bayesian estimation and quantitative model comparison by making parametric assumptions for the shape of neural tuning (a circular Normal distribution). The second makes no assumptions about the shape of neural tuning (i.e., a non-parametric analysis), providing a qualitative check of whether similar conclusions are reached. Critically, both techniques are needed, with the parametric model not only providing statistical model comparison, but also indicating the expected qualitative data patterns for the non-parametric check. As applied to BOLD data collected from the primary visual cortex of human participants viewing oriented gratings at high versus low contrast, the techniques correctly implicated multiplicative scaling, despite working with only voxel-level data.

## Parametric analysis: Linking neural tuning to voxel tuning
We first use INTM to build a parametric model of voxel tuning for inferring how changes in stimulus contrast modulate neural tuning. The resulting model provides a concrete demonstration of the ways in which different kinds of neural tuning modulation should produce different behaviors at the level of the voxel.

The parametric modeling approach involves assuming specific forms of the neural tuning functions, and further specifying the contribution of these tuning functions to a voxel's activity with a continuous distribution. For application to orientation preferences, which lie on a circular measurement scale, the contributions are assumed to follow a circular Normal distribution (Fig. 2). The circular Normal distribution resembles a typical Normal distribution but is periodic, thereby accommodating the circular nature of orientations. The distribution's density function has a period of $2\pi$, but orientations have a period of $\pi$, so the values of the orientations used in an experimental design are doubled when applying the model. For an orientation, $r$, the density function is

$$f(r|\phi, \kappa_1) = \frac{\exp(\kappa_1 \cos(r - \phi))}{2\pi I_0(\kappa_1)} \tag{1}$$

where $\kappa_1$ is a concentration parameter, $\phi$ is the orientation of the peak value of the function, and $I_0(\cdot)$ is the modified Bessel function of the first kind, of order 0. The numerator of Eq. 1 produces a periodic bell shape peaking at $\phi$, with the concentration parameter determining its "sharpness".

In the parametric model, there are two uses of the circular Normal distribution. First, Eq. 1 describes the relative contribution to a voxel's activity of different subpopulations of neurons that prefer different orientations, and so we refer to it as a "weight distribution". With a flat weight distribution (low $\kappa_1$), the neurons within a voxel are in equal proportion (resulting in a complete absence of voxel tuning even if the contributing neurons are well-tuned). With a peaked weight distribution (high $\kappa_1$), most of a voxel's activity is driven by neurons tuned to orientations around $\phi$ (resulting in sharp voxel tuning if the contributing neurons are also well-tuned).

A circular Normal density function fits single cell orientation tuning functions adequately[23] and so, in INTM, a second circular Normal distribution function is used to specify the individual neural tuning functions (Eq. 2). Although this equation relies on the distribution function, the interpretation is not a probability density but instead a description of neurons' responses to

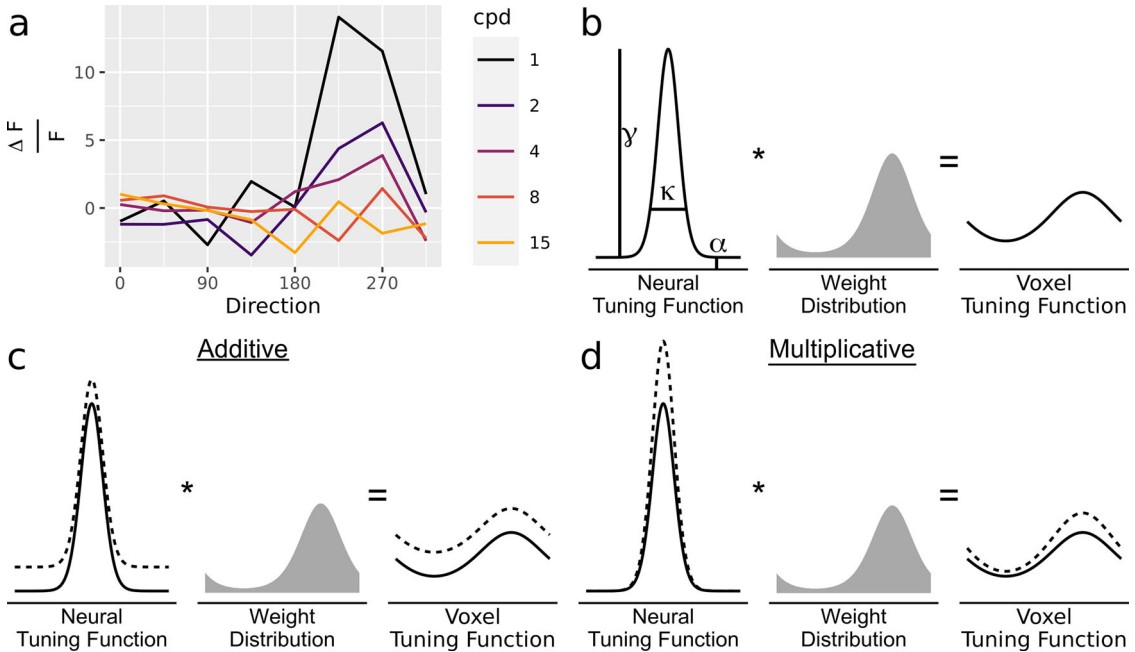

**Fig. 1 Modulations in neural tuning may manifest in voxel activity. a** Modulation of neural tuning. In this neuron (mouse primary visual cortex)[3], direction tuning interacts multiplicatively with spatial frequency (cpd: cycles per degree; $\Delta F/F$: average change in fluorescence). **b** Convolution ($*$) of neural tuning functions with a weight distribution leads to voxel tuning functions. The modeling procedure of INTM assumes that the observed voxel tuning functions reflect the shape of the neural tuning functions as applied across the distribution of preferred orientations (the weight distribution) within each voxel. Compare with Eq. 3. **c** An additive shift in the neural tuning function causes an additive shift in the voxel tuning function. **d** Multiplicative gain in the neural tuning function causes multiplicative gain in the voxel tuning function. Compare **c** and **d** with Equation 4.

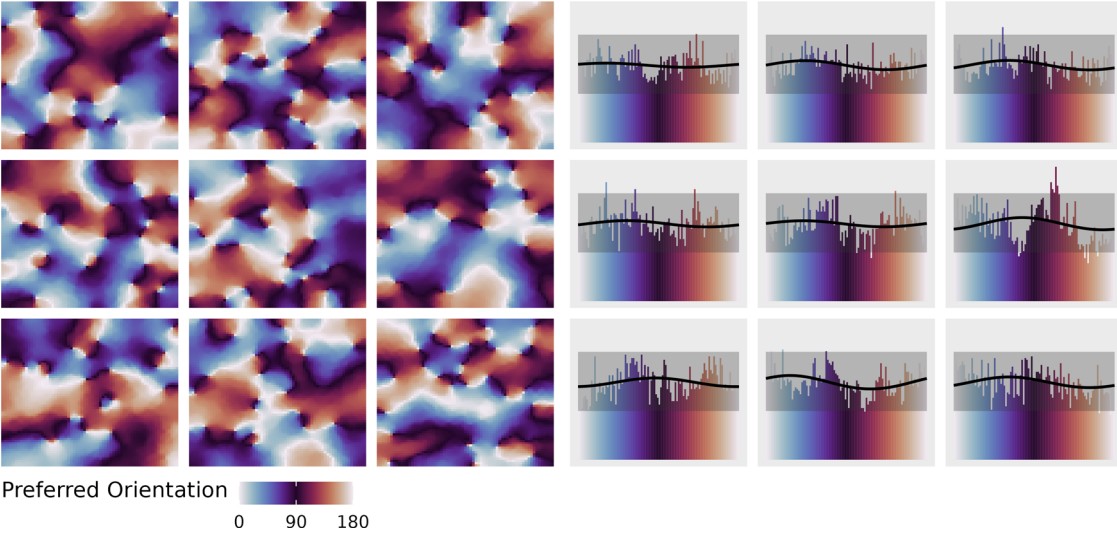

**Fig. 2 Orientation preferences of simulated voxels can be approximated with unimodal distributions. Left:** A Kohonen network was used to simulate orientation preference maps. Each panel represents a separate voxel. **Right:** Weight distribution for each voxel. The histograms represent the tallies of each preferred orientation in the left panels. The black line is the best fitting (maximum likelihood estimation) circular Normal density function (Eq. 1). The horizontal gray (shaded) ribbons give the 99% confidence interval for a flat distribution (i.e., the range of heights that we would expect when a simulated voxel contains an equal number of neurons tuned to all orientations). Given the amount of cortex sampled by each stimulated voxel, most weight distributions tend to be flat, but the fitted von Mises density function can capture individual modes.

orientations. In this equation, the concentration parameter indicates the sharpness of a single cell's tuning. To account for a neuron's baseline activity and responsiveness, we included additional parameters. The baseline activity parameter reflects the activity of a neuron in the absence of stimulation and is implemented as an additive offset, $\alpha$. The parameter for responsiveness models how well a neuron responds to stimulation and is implemented as a gain parameter, $\gamma$. With the addition of these baseline activity and responsiveness parameters, Eq. 2 specifies the neural tuning function, $NTF(\cdot)$ for orientation of a presented stimulus relative to that neuron's preferred orientation.

$$NTF(r|\kappa_2, \gamma, \alpha) = \alpha + \gamma f(r|0, \kappa_2) \qquad (2)$$

Within each voxel (but not across voxels), all neural tuning functions are assumed to take on the same circular Normal parameter values. The voxel tuning function is obtained by convolving Eqs. 1 and 2 (Fig. 1b). This results in Eq. 3, the voxel tuning function, $VTF(\cdot)$, in a baseline condition (e.g., low-contrast stimuli). It is critical to note that the choice of baseline condition is arbitrary (e.g., the high-contrast condition could be labeled as the baseline) considering that the model is applied to both conditions conjointly. The only difference in terms of which condition is labeled as baseline is whether the modulation parameters are increases or decreases.

$$VTF_{baseline}\left(r|\kappa_v, \gamma_v, \alpha_v, \phi_v\right) = \alpha_v + \gamma_v f\left(r|\phi_v, \kappa_v\right) \quad (3)$$

The subscript $v$ indicates parameters that are estimated separately for each voxel. The concentration parameter, $\kappa_v$, is equal to neither $\kappa_1$ nor $\kappa_2$, but is a function of both[24]. Thus, the model does not distinguish between voxels containing a concentrated distribution of orientation-tuned neurons each of which has diffuse tuning (high $\kappa_1$ and low $\kappa_2$), and voxels containing a diffuse distribution of orientation-tuned neurons each with concentrated tuning (low $\kappa_1$ and high $\kappa_2$). The key insight behind INTM is the realization that there is no need to differentiate between such possibilities, if the measurement goal is identification of the qualitative form of neural tuning modulation rather than precise specification of neural tuning functions.

Within the INTM framework, each form of tuning modulation is instantiated by incorporating additional parameters that modulate one or more parameters in the baseline voxel tuning function (Eq. 3, Fig. 1c, d). In this validation test case as applied to visual contrast manipulations, we considered two forms of tuning modulation that are capable of producing an average increase in neural activity: multiplicative gain and additive shift. Other forms of neural tuning modulation, such as bandwidth changes and tuning preference shifts, could be implemented within the INTM framework, but these forms were not included in the candidate set of models for this test case because they do not change average neural activity (i.e., they are not viable models for the neural modulation that underlies changes in visual contrast).

In this test case, multiplicative gain is known to be "ground truth"—that is, changes in visual contrast induce multiplicative gain in single-neuron tuning functions[19–21]. However, in seeming contradiction to multiplicative modulation, an examination of raw voxel tuning functions suggests an additive increase with increases in visual contrast (Fig. 4a, b). Nevertheless, averages can be misleading, and a formal model comparison between additive shift and multiplicative gain is required that considers each voxel separately, particularly when considering that many of the voxels contributing to the average exhibit poor tuning. For completely untuned voxels, additive shift and multiplicative gain are indistinguishable and so the average will necessarily look additive. If INTM works, it should leverage differences between voxels (i.e., capitalize on the small proportion of well-tuned voxels), to reach the conclusion that multiplicative gain is the more likely form of tuning modulation despite what the average BOLD signal appears to show.

To model a change in multiplicative gain, we included a voxel-specific gain multiplier parameter, $g_v$ (Eq. 4a). To model an additive shift change, we included another, voxel-specific additive shift parameter, $a_v$ (Eq. 4b). In summary, Eq. 3 captures the situation for low contrast in both models, whereas either Eq. 4a or 4b captures the situation for high contrast.

$$Multiplicative: VTF_{modulated}\left(r|\kappa_v, \gamma_v, \alpha_v, \phi_v, g_v\right) \\ = \alpha_v + g_v\gamma_v f\left(r|\phi_v, \kappa_v\right) \quad (4a)$$

$$Additive: VTF_{modulated}\left(r|\kappa_v, \gamma_v, \alpha_v, \phi_v, a_v\right) \\ = a_v + \alpha_v + \gamma_v f\left(r|\phi_v, \kappa_v\right) \quad (4b)$$

For other applications, shifts in orientation preferences could be modeled by allowing $\phi_v$ to change across conditions, and a change in concentration could be modeled by allowing $\kappa_v$ to change across conditions. In the current dataset, these forms were not pursued because neither allows contrast to alter the average activity of a voxel across orientations—an effect that was clearly present in the empirical data (Fig. 4a)—and so these forms could be rejected without explicit modeling.

Equations 3, 4a, and 4b model voxel activity at each tested orientation, at each level of contrast. However, there is variability in voxel activation across runs, and this variability varies across voxels (some voxels are noisier than others, Supplementary Fig. 2). These sources of variability were included in the model, and all voxel-specific parameters were estimated hierarchically (Supplementary Methods). Hierarchical estimation of voxel tuning functions combines information across voxels, allowing model comparison to automatically down-weight the noisier voxels and magnify the well-tuned voxels[25].

The parametric model assumes that all neurons within a voxel share a common shape for their tuning function (homogeneity of tuning shape), which is unlikely to hold strictly true. However, some degree of tuning shape heterogeneity should not invalidate inferences drawn about the form of tuning modulation, provided that tuning function shape does not differ systematically as a function of the orientation preference of the neurons. That is, there could exist a range of tuning function shapes, but if that heterogeneity occurs to the same extent for neurons centered on all preferred orientations, then it should not lead to scenarios where one form of modulation mimics another at the voxel level. Nevertheless, to assess whether such heterogeneity posed a problem when adjudicating between the forms of tuning modulation, we also developed a non-parametric check that does not make this assumption. If the non-parametric check supports the same qualitative conclusion as the parametric model, this suggests that the homogeneity of tuning shape assumption was adequate.

## Model recovery

INTM judges the relative plausibility of each form of neural tuning modulation (e.g., Eq. 4a versus 4b) with model comparison based on empirical data. But under what circumstances should model comparison be trusted? To assess the extent to which inherent properties of the models (e.g., their flexibility) allow reliable model comparison, we assessed model recovery via two kinds of simulation: data-informed and data-uninformed[26]. Model recovery entails producing simulated data from each model, asking whether the model that produced the simulated data is better able to capture its own data than competitor models. If a model is overly flexible, then it might provide a better explanation of data generated by other models than do the generating models themselves.

Data-uninformed model recovery asks whether the model comparison technique is well-calibrated to the relative flexibility of each model in general. First, synthetic data were generated with each model using "weakly informative" priors (Supplementary Methods). These priors were not designed to reflect precise experimental knowledge but instead simply provide a rough scale for each parameter individually. Next, the models were applied to these synthetic data, keeping in mind the "ground truth", data-generating model. The model comparison technique has built-in adjustments for flexibility by approximating the ability of each model to predict held-out data (i.e., cross-validation). However, this approximation might be inadequate, providing too much or

too little penalty for model flexibility. Data-uninformed recovery asks whether this model comparison is operating as it should in general.

There is no guarantee that a specific real dataset will be diagnostic regarding a particular model comparison. For instance, if none of the voxels is sufficiently well-tuned, then the two forms of tuning modulation will make the same prediction. Similarly, if the manipulation of visual contrast is too weak, it would not be possible to reach a reliable conclusion. Whether the real dataset is sufficient for differentiating between the candidate models is assessed using data-informed model recovery by repeating the recovery process using what has been learned about the parameters of the models from the empirical data.

Whereas data-uninformed model recovery assesses the fairness of model comparison across a wide range of individual parameters that are plausible given the prior distribution, data-informed recovery generates synthetic data by sampling from the posterior distribution obtained by estimating the models with empirical data. Generating synthetic data that are consistent with real data constrains the parameters (e.g., tuning width and modulation) to a much narrower and more realistic range of values. Indeed, the behavior from the posterior parameter values closely matches the weak tuning effects seen in the real data at the level of separate voxels (e.g., Supplementary Fig. 5). Because the data-uninformed model recovery has priors that are only weakly informative, it can allow extreme parameter values for estimated beta-weights, and hence implausibly sharp tuning functions; this increases the chance that each generating model will produce a unique signature pattern in the data, making model recovery likely to succeed. By contrast, the more constrained data-informed model recovery presents more of a challenge to model recovery, providing greater reassurance in the case that it succeeds.

## Non-parametric check: An orthogonal regression slope analysis

The parametric model assumed normal functions for the NTFs and weight distributions and additionally assumed homogeneity of NTF shape within a voxel. These assumptions will often be violated, but such violations might not pose a problem for model comparison, particularly if the violations of these assumptions are uniformly applied across the stimulus dimension of interest. Nevertheless, as a complement to parametric model comparison, we developed a more qualitative non-parametric check of the empirical data. If this check supports the same conclusion as the parametric model, this suggests that any violations of parametric assumptions were insufficient to alter the theoretical conclusions.

The parametric model is crucial to the development of the non-parametric check and the specific parametric models under consideration should first be used to generate predictions for the check. The consequences of parametric assumptions are then considered in light of these predictions. In the case of a comparison between multiplicative gain versus additive shift, these two forms of tuning modulation can be distinguished using voxel-wise regression; if the modulation is multiplicative, a within-voxel plot comparing high- versus low-contrast should have a slope greater than 1, whereas if the tuning modulation is additive, this plot should reveal a slope equal to 1. Furthermore, these predictions do not rely on parametric assumptions. For instance, if a voxel is multimodal, preferring more than one orientation, this would serve only to rearrange the order of the orientations in the regression plot, but the qualitative distinction between a slope of 1 versus a slope greater than 1 would still

map onto additive shift versus multiplicative gain. The generality of the slope test across all shapes of tuning function and all shapes of weight distribution is mathematically proved in Supplementary Methods Eqs. 1–4.

The key difference between these two forms of tuning modulations is that the multiplicative model allows the effect of contrast to vary by orientation whereas the additive model does not. Additive modulation corresponds to an increase in neural activity at all orientations. Thus, regardless of the weight distribution and regardless of the shape of the neural tuning functions, additive tuning modulation causes the low-contrast tuning function to shift upwards uniformly across orientations (Fig. 3, top left). Hence, a scatterplot of the voxel's response to high-contrast stimuli against its response to low-contrast stimuli has a slope of 1 (Fig. 3, top right). In contrast, multiplicative tuning modulation corresponds to a greater increase in neural activity at the most preferred orientations (Fig. 3, bottom left), and a scatterplot of a voxel's response to high versus low-contrast stimuli has a slope larger than 1 (Fig. 3, bottom right). Therefore, the models can be differentiated by plotting high-contrast activity against low-contrast activity and calculating the slope of the best fitting line (Fig. 3, right). A slope of one implies additive shift, but a slope greater than one implies multiplicative gain (see also Supplementary Methods). The only assumption made in this analysis is that the magnitude of the additive shift or multiplicative gain is the same for all neurons contributing to a particular voxel or, more specifically, that if the magnitude of modulation does vary across neurons, it does not vary systematically with neurons' preferred orientation (see Supplementary Information).

Analyzing the slope is not necessarily straightforward. First, the slope between high- and low-contrast cannot be determined using standard linear regression (i.e., estimating the line that minimizes the ordinary least squares error—the squared vertical distance between the line and the data). Standard linear regression estimates a relation between two sets of observations, such as the activity at low and high contrast. But when *both* of those observations are corrupted by noise, the estimated relationship is "diluted" and biased to zero[27]. To mitigate this dilution, the slope was estimated using *orthogonal* regression[27], by finding the line that minimizes the squared Euclidean distances from the data (Methods, Supplementary Methods). Second, the slope analysis cannot be done across voxels because there may be covariance between a voxel's low contrast (baseline) activation and the effect of contrast on the voxel's activity (e.g., voxels with a larger response in the low-contrast condition may receive a larger contrast effect, and this tendency would bias an across-voxel analysis toward a slope greater than 1). To mitigate this confound, slopes were determined separately for every voxel, followed by an analysis of the distribution of slopes. Finally, even a within-voxel slope analysis produces results that are difficult to interpret because most voxels are only weakly tuned, responding to all orientations almost equivalently. For those voxels, the data for the slope analysis will be an uncorrelated cloud of points, and an orthogonal regression for a cloud of points can take on nearly any slope value, including positive or negative infinity for the case of vertical slopes. Therefore, we report, not the slope of the line, but instead the angle of the slope.

In using INTM, we recommend first developing and comparing parametric models of how stimuli give rise to voxel activity during manipulations of cognitive or perceptual state, and then, in addition, to consider ways of relaxing those parametric assumptions to provide a non-parametric check of the model comparison results, as with the slope test for comparisons of multiplicative versus additive changes.

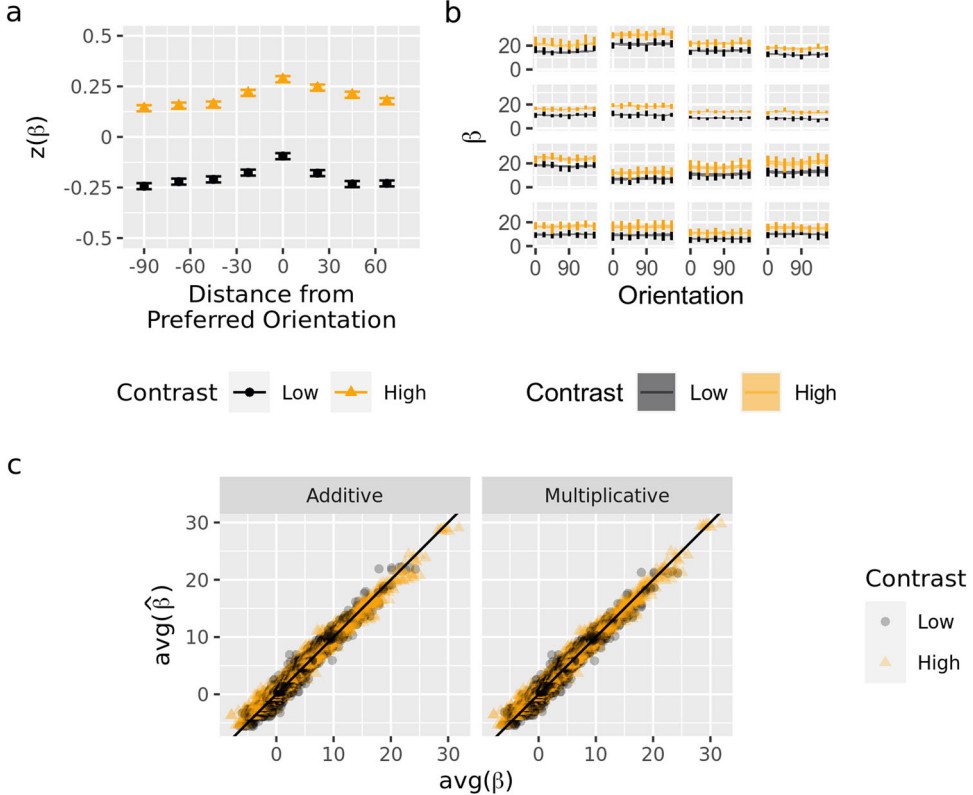

**Fig. 3 Average voxel tuning functions visually suggest an additive shift even though the multiplicative model is better in its posterior predictions of individual voxel data. a** Average voxel tuning at low and high contrast using binning to average (see Methods). Error bars give 95% confidence interval for the mean of the 1010 V1 voxels across seven participants. **b** Examples of individual voxel BOLD data in comparison to posterior predictions of the multiplicative gain model. The panels show the 16 voxels with the largest average difference in activity across levels of contrast. Vertical error bars span 95% confidence intervals (within-subjects), and the width of the lines indicate the 95% highest density interval of the posterior predictive distribution for the multiplicative model (additive model looks similar). **c** Overall comparison between each model ($\hat{\beta}$) and observed data ($\beta$). Each point corresponds to a single voxel's average activity at a given orientation at a given level of contrast. Panels show the average of the posterior for predictions from either the additive or multiplicative model.

## Results

**Binning voxels to plot average voxel tuning**. Voxel tuning functions were first analyzed with a standard approach of "averaging" across circularly aligned voxels (Fig. 4)[10], which is an appropriate technique if all voxels have similarly shaped tuning functions. This "binning" alignment technique appears to reveal an additive shift with contrast, in contrast to what is known from electrophysiology. This apparent additive shift is caused by the inclusion of many voxels with poor orientation tuning. When the BOLD response is aggregated across a many such "undiagnostic" voxels, the true form of tuning modulation is obscured (i.e., it appears additive rather than multiplicative). One approach to mitigating this variability would be to adopt experiment-specific thresholds, excluding some subset of "relatively" untuned or unresponsive voxels. However, it would be difficult to justify such thresholding for each experiment, and more importantly excluding voxels throws away information (e.g., about the average noise in voxels' responses). The two techniques of INTM avoid thresholding, even while using the well-tuned "diagnostic" voxels to identify the most likely form of tuning modulation.

**Data-uninformed model recovery**. To evaluate the ability of the INTM parametric model to recover both kinds of modulation, we conducted model recovery by simulating datasets from the prior distribution for each candidate model, applying both models to both sets of generated data, and comparing the results using an approximation to cross-validation[26,28] (see Methods). The

models did not exhibit mimicry: model comparison picked the true data-generating model in every simulated dataset, providing strong evidence that the technique recovers the appropriate class of modulation.

**Application of the parametric model to the observed data**. We next asked whether the parametric INTM analysis recovers the correct form of tuning modulation when applied to the observed fMRI data. Specifically, it is known from electrophysiology that the tuning functions of orientation-sensitive neurons undergo multiplicative tuning modulation with an increase in stimulus contrast[19–21]. Given this knowledge, model comparison on the collected BOLD data ought to favor the multiplicative model. Visual comparison of the posterior predictive distribution from both models revealed that both models largely captured individual voxel tuning functions adequately (Fig. 4b, c). The models were then compared to each other with cross-validation[28]. The multiplicative model provided the best account of the data: a difference of 14.5 standard errors, in units of expected log pointwise-predictive density[28]. A significance threshold (e.g., $\alpha = 0.05$ or 1.65 standard errors) would suggest that this difference in predictive ability is poorly accounted for by a null model in which the two forms of tuning modulation are equally predictive.

**Data-informed model recovery**. Exploiting what was learned about the plausible parameter values by observing the data, we further probed the validity of model comparison through

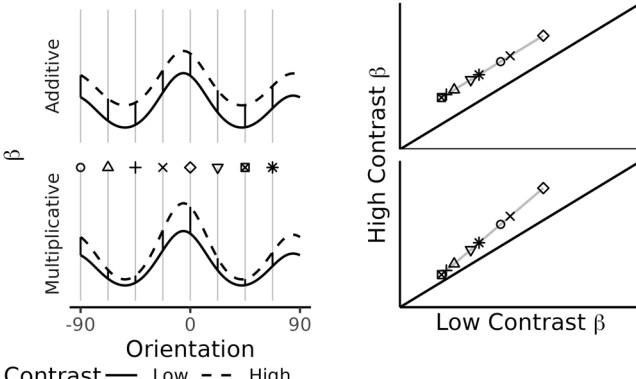

**Fig. 4 Plotting a voxel's response to high versus low contrast orientation uncovers the form of tuning modulation. Left:** Simulated voxel tuning functions in which higher levels of contrast induce either an additive (top) or multiplicative (bottom) tuning modulation. The eight vertical lines are eight hypothetical orientations at which these voxel tuning functions might be probed, which would produce eight responses per level of contrast, High vs. Low. Note that, whereas the parametric model assumes that the weight distribution and neural tuning function follow specific shapes (e.g., unimodal distributions), the slope test does not (e.g., the voxel tuning function can be bimodal, as displayed). **Right:** The two kinds of neural tuning modulation reveal different signatures when the responses to high contrast stimuli are plotted against the responses to low-contrast stimuli. The diagonal line corresponds to no effect of contrast. A line drawn through the points produced by the additive model necessarily has a slope equal to 1 (top); under this form of modulation, the effect of contrast does not depend on the orientation. A line drawn through the points produced by the multiplicative model necessarily has a slope greater than 1 (bottom); under this form of modulation, the effect of contrast is largest at those orientations that are closest to the voxel's preferred orientation.

data-informed model recovery[26,28]. Synthetic datasets were generated from the posterior distributions of the multiplicative and additive models. Next, the multiplicative and additive models were applied to each of these simulated datasets. All datasets were best accounted for by the model whose posterior was used to generate the data even though, as seen in Fig. 4, the differences between the models were very subtle.

**Orthogonal regression.** The parametric model correctly indicated that an increase in visual contrast caused an increase in multiplicative gain, but this result relied on assumptions of circular Normal distributions for the neural tuning function and distribution of weights for each voxel, as well as homogeneity of tuning shape for each voxel. However, we determined that the two candidate forms of tuning modulation should be distinguishable based on what they predict for the slope of a plot of high-contrast activity against low-contrast activity (Fig. 3, Introduction, Supplementary Methods), without relying on these parametric assumptions and the assumed homogeneity of tuning shape. As predicted by multiplicative tuning modulation, this non-parametric orthogonal regression slope analysis revealed slopes larger than 1, corresponding to slope angles greater than 45 degrees (Fig. 5). Figure 5b shows the distribution of within-voxel slopes, as estimated with orthogonal regression of the high- on the low-contrast activity (Introduction, Methods). The median slope had an angle greater than 45° (i.e., slope of 1). This tendency for slopes to be larger than 1, as predicted by the multiplicative model, was further supported by a hierarchical Bayesian estimation of the slopes (Supplementary Methods).

## Discussion

Many theories of cognition predict specific kinds of neural tuning modulation with different manipulations e.g.,[29,30]. For instance, attention researchers do not only consider neural responses when people are in a state of high attention; this state is compared and contrasted to one of low attention, and theories of attention are informed by this comparison e.g.,[31,32]. However, these predictions for tuning modulation – the manner in which neural tuning functions change – are rarely tested in humans because there are no non-invasive techniques for recording from individual neurons. Tuning functions, and modulation of tuning functions, can be measured non-invasively using fMRI, but the response of a voxel reflects hundreds of thousands of neurons. Thus, voxel tuning functions have been interpreted as population-level properties of stimulus representations in the brain, population-level responses that cannot be used to infer properties of the underlying, individual neurons[10,12–14].

Rather than attempting a quantitative measurement of neural-level tuning from fMRI data, we developed a framework, INTM, for identifying the functional form of neural-level tuning modulation. A key component underlying the INTM framework is a modeled link between the neural and voxel tuning functions (Fig. 1b–d). This link justifies inferences of neural tuning modulation from voxel tuning modulation; even though the BOLD signal reflects the aggregated activity of many neurons, different forms of modulation can produce identifiable signatures in activity of the voxels (i.e., Eq. 4a vs 4b, Fig. 3), thereby allowing identification of the most likely form of modulation by applying each model to the BOLD data. We tested that link and validated INTM with a test case in which the neural-level "ground truth" modulation was known to be multiplicative gain[19–21]. This test case was particularly challenging because the average voxel results appeared to indicate a clear additive shift rather than multiplicative gain. We used INTM to design two techniques: 1) parametric assumptions of tuning function shapes and hierarchical Bayesian estimation of the models followed by formal model comparison; and 2) a qualitative, non-parametric check using orthogonal regression, without relying on parametric shape assumptions. Both techniques of the INTM framework correctly implicated multiplicative gain of neural tuning by leveraging subtle effects that were more apparent in some voxels than others.

Several techniques have previously been developed to peer inside the voxel, but none of them statistically compares alternative theories of tuning modulation using a model that explicitly links neural tuning with voxel tuning. For example, biophysical models specify how the BOLD signal arises from the often non-linear coupling between neural activity and vasculature[33,34]. But, despite this biological realism, these models have not yet been used to examine changes in neural tuning functions. In contrast to biophysical models, encoding models of fMRI data delve inside the voxel by modeling how components comprising a voxel (e.g., sub-voxel "channels") transform stimuli into voxels' activity[5,15–17], and, promisingly, enable researchers to use the BOLD signal to uncover features of neural tuning[18]. Encoding models can be applied to multiple conditions – for example through simulation[35], or through fitting to data and inverting[36] – thereby estimating how the channels are modulated by an experimental manipulation. Inverting an encoding model uses it as a decoder, estimating channel responses for other stimuli[15,16,37] or estimating how channels are modulated in other conditions[38]. But this use of inverted encoding models requires hard-wired (and arbitrary) assumptions about the shape of the channel responses in baseline conditions[39,40]. These assumptions are even more restrictive than those made by the parametric model in INTM considering that they not only assume a specific class of shape (e.g., a circular Normal), but they require adopting

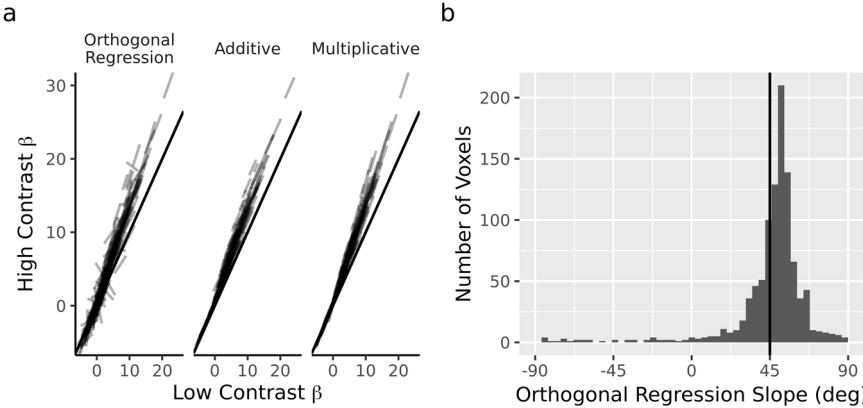

**Fig. 5 Non-parametric check of model comparison using orthogonal regression analysis. a** Orthogonal regression on data and model predictions. Each line corresponds to a single voxel. Most voxels are weakly responsive at each level of contrast and are only weakly influenced by the experimental manipulation (i.e., lines cluster in the lower left corner, close to the diagonal). Across voxels (i.e., comparing separate lines), both models capture larger visual contrast effects for more responsive voxels: predictions lying further rightward along the x-axis are located further away from the diagonal. However, only the multiplicative model captures the observed interaction between orientation and contrast within voxels. Compare to Fig. 3. **b** Distribution of within-voxel slopes. As allowed by multiplicative but not additive modulation, the distribution of slopes is shifted above 45° (i.e., slope of 1).

a specific, user-selected parameter value for that shape (e.g., a particular value for the width of the circular Normal). These assumptions mean that the inferred channel modulation cannot be taken to reflect the modulations in underlying neural functions that would be measured with electrophysiology, as users of such models readily acknowledge[14,39]. As a result, inverted encoding models have been used to pursue very different goals than the aim of the present study[14] (i.e., understanding population- rather than neural-level representations). Beyond inverted encoding models, other researchers have compared the qualitative predictions of different forms of tuning modulation[35,41,42], analogous to the logic underlying INTM's non-parametric check. However, these techniques neither fit the alternative encoding models to empirical data, nor account for model flexibility.

These previous attempts to peer inside voxels are combined in INTM. Like biophysical models, INTM connects the latent neural activity to the observed BOLD. It does so by assuming a linear coupling that is justified given our experimental design and the goals of our modeling procedure (we discuss potential violations of this assumption in more detail below)[43–46]. But unlike these biophysical models, INTM uses an encoding model (Fig. 1b–d, Eqs. 3, 4a, 4b). Like inverted encoding models, the parametric analysis in INTM assumes a specific class of neural tuning shape (circular Normal). But unlike inverted encoding models, it avoids the need to assume specific parameter values for the shape (e.g., tuning width). The non-parametric slope analysis in INTM takes things a step further by avoiding the need to assume a specific class of neural tuning shape. In applying the framework to data, we used a quantitative measure of the predictive abilities of each model (a form of cross-validation), a measure that is sensitive to both the flexibility of each model and the signal-to-noise ratio of the data.

INTM is a general framework for inferring neural tuning modulation, but the two techniques developed in this report have a crucial feature that ought to be retained in further applications of the framework: analyses were within-voxel. As previously reported for voxel-wise orientation tuning[6], most of the voxels in our study were poorly tuned, and they were additionally only weakly affected by stimulus contrast. Although we restricted the analyses to only those voxels whose population receptive fields overlapped with the stimulus (see Methods), most voxels were only weakly responsive (e.g., falling near the origin in Fig. 5a) and were minimally affected by visual contrast (e.g., lying near the

diagonal in Fig. 5a). These voxel differences and other sources of voxel variation—including partial voluming[47], proximity of each voxel to blood vessels[48], the responsiveness of neurons within each voxel, etc. –are present in any fMRI experiment. Analyses that are insensitive to weak signals risk conflating different forms of tuning modulation, particularly when the noise differs across conditions (Supplementary Methods—Non-Parametric Check)[22]. In the current case, because most voxels exhibited nearly flat tuning, this altered the appearance of the average voxel tuning function (Fig. 4a), incorrectly suggesting that changing visual contrast causes an additive shift. This was because an additive shift is equivalent to multiplicative gain for a voxel with poor tuning. Both INTM techniques avoided this problem by assessing the voxels individually, identifying (and exploiting) the voxels that exhibited the unique combination of being well-tuned and possessing a strong visual contrast effect. Voxels with this combination of behaviors were more diagnostic for the comparison between different forms of tuning modulation.

The techniques developed here were tailored to uncover changes in orientation tuning caused by stimulus contrast, relying on assumptions that will not be appropriate in all studies. We highlight these assumptions here and use them to clarify the distinction between the framework more generally versus specific applications of the framework. First, the models assumed that voxel tuning functions are a linear combination of the activity of the underlying neural tuning functions. This assumption is reasonable in some experiments[43,44], but the relationship between neural firing rate and BOLD signal is not perfectly linear[46,49]. Second, the techniques implicitly assumed, through the general linear model used to estimate voxel activity, a hemodynamic response function that is shared across all voxels in all participants. Although canonical, that assumption is erroneous e.g.,[50]. Next, the parametric analysis approximated the distribution of neurons tuned to orientations within each voxel with a circular Normal distribution, which is unimodal and periodic. This approximation provided a convenient formula for deriving the voxel tuning function, and the resulting function matched the data (e.g., individual voxel tuning functions in Fig. 4b appear unimodal). However, this assumption may not hold for other types of stimuli, other brain regions, or for other scan parameters (e.g., an aperiodic stimulus like pitch would require an aperiodic tuning function).

Violations of these assumptions may bias model comparison in some cases but, fortunately, none of the assumptions is intrinsic to the framework, and there are several ways of relaxing each of them. For example, more complex weight distributions such as multimodal functions could be used in the parametric model. Alternatively, we have shown that assumptions about the weight distributions can be bypassed by a technique like the non-parametric slope analysis. As an another example, although nonlinear relationships between the neural firing rate and the BOLD signal have been documented, multiple mathematical accounts have been proposed to explain these nonlinearities[49,51,52], and these relationships could be incorporated into INTM.

Despite its generality, INTM has two limitations that must be plainly acknowledged. The first is that INTM requires that the models under consideration make distinct predictions at the level of the voxel. For instance, the parametric and non-parametric INTM techniques both assume that the magnitude of tuning modulation is the same for all neurons that contribute to the voxel response. These techniques may be robust to relaxing this assumption if modulation magnitude differences apply uniformly across the weight distribution that maps neural preferences onto voxel preferences. But if the magnitude of modulation varies systematically, the models might no longer make distinct predictions at the level of the voxel. For example, if the magnitude of the additive shift is larger for neurons that prefer the same orientation as the voxel, this unlikely circumstance could produce a change in the voxel tuning function that is indistinguishable from a constant magnitude of multiplicative gain. The inability to discriminate these two scenarios exemplifies the fundamental inverse problem that INTM cannot solve: INTM cannot determine the parameters of tuning functions for individual neurons (indeed, it is not designed to). Instead, INTM operates upon distributions of neurons (grouped by, e.g., region-of-interest, voxel, feature preferences), and so it can compare only models whose distributions of tuning functions change in ways that are distinct. With the example of an additive shift whose magnitude happens to covary with the weight distribution, this might occur for some voxels, but is unlikely to occur with most voxels, demonstrating the need to use all of the voxels in the model comparison.

The second key limitation is that INTM can differentiate only between models that are formally included in the model comparison process. If the "true" form of tuning modulation is not included in the set of possibilities, the results may mislead researchers into conflating the winning form of modulation with the true form. For instance, in the present study, the technique contrasted tuning modulation models in which just one form of modulation occurred but did not consider more complex situations in which multiple forms of tuning modulation could have occurred. For example, it is known from electrophysiology that with extended adaptation, orientation tuning functions can both widen and shift their preferred orientation[53]. Since that more complex model, in which tuning widened and shifted within every trial, was not included in model comparison, INTM is silent about whether that effect occurred. Note that such combinations of tuning modulation could be included in the model comparison process, but when attempting to adjudicate between complex models it would be critical to use model recovery simulations, specifically data-informed recovery, to determine if the collected data were sufficiently constraining.

In summary, using a simple test case—modulation of orientation tuning by stimulus contrast—we presented and validated the INTM method for identifying the form of neural tuning modulation from a BOLD dataset. The method is applicable to a broad range of domains and manipulations. The main insight of the framework is that there are reasonable assumptions, which can be made in many neuroimaging studies, that enable inference about neural tuning modulation. The main requirement for using the technique is an experimental paradigm that: (1) produces measurable voxel tuning by testing the BOLD response at different levels of a single stimulus dimension (e.g., direction of motion, color, pitch), and (2) includes some cognitive manipulation of interest (e.g., with and without attention, before and after perceptual learning, complex versus simple stimuli) that modulates the voxel tuning from one condition to another. Now that INTM has been validated, it can be used to study modulations of neural tuning where electrophysiology has not yet provided an answer (e.g., studies of tuning modulation underlying tasks that cannot be taught to animals).

## Methods

**Participants.** Seven participants (22–31 years old; 3 female, 2 did not report) completed three sessions for monetary compensation ($50 per 2 h session). All participants had normal or corrected-to-normal vision. One additional participant completed a single session but exhibited substantial motion; their data were excluded. The procedure was approved by the University of Massachusetts Institutional Review Board.

**Behavioral stimulation and recording.** Stimuli were presented to participants with a gamma corrected 32" LCD monitor at 120 Hz refresh rate (Cambridge Research Systems). The experiment was designed using the Psychophysics toolbox (Version 3.0.14)[54] and custom MATLAB code (2018b, MathWorks). Behavioral responses were collected with a button box (Current Design). Eye-tracking data were recorded at a rate of 1000 Hz with the Eyelink 1000 Plus system on a long range mount (SR Research), controlled using the Eyelink Toolbox extension to Psychtoolbox[55]. Due to a technical error, the button presses of one participant were not recorded.

For the main experiment, participants completed 18 functional runs across three sessions. During each run, oriented grayscale gratings were presented twice at each of two levels of contrast (eight orientations at 50% or 100% Michelson contrast in all runs for six out of seven participants, eight orientations at 20% or 80% Michelson contrast in 12 runs for one participant, and seven orientations at 20% or 80% Michelson contrast in that participant's remaining 6 runs). Grating parameters replicated those of Rademaker et al.[56]. Gratings (spatial frequency of 2 cycles per degree) were masked with annuli (1.2° inner and 7° outer radii). The annulus edges were smoothed with an isotropic 2D Gaussian kernel (1° kernel, 0.5° standard deviation). Throughout each run, a magenta fixation dot was presented in the center of the screen (0.2°, RGB: 0.7843, 0, 0.8886).

In each trial, a counterphasing (5 Hz) grating was presented for five seconds. In the middle three seconds of each trial, the spatial frequency of the grating either increased or decreased (1 cycle per degree) for 200 ms. Participants were instructed to indicate via button press the direction of change as soon as they noticed it. Per run, gratings were presented at multiple orientations, twice at each combination of orientation and contrast. In most runs, there were eight orientations, but in one session of one participant (totaling six runs), only seven orientations were presented. Inter-stimulus intervals ranged from 8000 to 12000 ms in steps of 200 ms. A five-second fixation period preceded the first trial, and a fifteen-second fixation period succeeded the final trial. Each run lasted 490 s.

We mapped a circular area of the visual field, of radius 8° centered on a central fixation point. pRF mapping scans followed the protocol of Benson et al.[57]. Briefly, natural images[58] were overlaid on pink noise and viewed through a series of circular apertures (8° radius). Within one run per session, the apertures enabled view of either moving bars (2°) or rotating wedges (1/4 aperture) and rings that expanded and contracted (see stimulus software for details). In bar runs, a bar traversed the central region in cycles. During each cycle, the bar was visible for 28 s, followed by a 4 s blank period. The bar moved in one of eight directions (east, north, west, south, northeast, northwest, southwest, or southeast, in that order). A 16 s blank period preceded the first cycle, a 12 s blank period followed the fourth cycle, and there was a 16 s blank period at the end of all cycles (300 s in total).

In the second pRF scan, the apertures were either wedges that rotated clockwise or counterclockwise, or they were rings that expanded or contracted. These runs started with a 16 s blank period, followed by two, 32 s cycles of a counterclockwise rotating wedge, two 28 s of expanding rings (each followed by a 4 s blank period), two 32 s clockwise wedge rotations, and two 28 s cycles of contracting rings (followed by 4 and 26 s of blank, respectively). The total run time was 300 s.

Throughout the pRF scans, the color of a central fixation dot (0.3°) changed between black, white, and red. Participants were instructed to monitor the color of the fixation dot and press a button when the dot turned red. To help participants maintain fixation, a circular fixation grid was presented throughout.

**fMRI data acquisition**. MRI data were collected on a 3T Siemens Skyra scanner with a 64-channel head coil. In each of the three sessions we collected field-mapping scans, functional scans, and a T1-weighted anatomical scan (MPRAGE, FOV 256 × 256, 1 mm isotropic, TE 2.13 ms, Flip Angle 9°). The anatomical scan was used to align field-mapping and functional images parallel to the calcarine sulcus. Gradient recall echo scans estimated the magnetic field. The pRF and primary functional data were collected with the same scan parameters (TR 1000 ms, TE 31 ms, flip angle 64°, FOV 94 × 94, 2.2 mm isotropic, interleaved acquisition, no slice gap, Multiband Acceleration Factor 4). To aid alignment of functional and anatomical images, single-band reference images were collected before each functional run for all but three participants (TR 8000 ms, TE 65.4 ms, flip angle 90°, FOV 94 × 94, 2.2 mm isotropic, interleaved acquisition, no slice gap).

**MRI preprocessing**. Preprocessing of images was performed with *fMRIPrep* 1.4.0, which relies on *Nipype* 1.2.0 and *Nilearn* 0.5.2 (RRID:SCR 001362).

The T1-weighted (T1w) images were corrected for intensity non-uniformity (INU) with *N4BiasFieldCorrection*, distributed with ANTs 2.2.0. The T1w images were then skull-stripped with a *Nipype* implementation of the *antsBrainExtraction.sh* workflow (from ANTs), using OASIS30ANTs as target template. A T1w-reference map was computed after registration of the individual T1w images (after INU-correction) using *mri_robust_template* (FreeSurfer 6.0.1; RRID: SCR 001847). Brain surfaces were reconstructed using *recon-all*, and the brain mask estimated previously was refined with a custom variation of the method to reconcile ANTs- derived and FreeSurfer-derived segmentations of the cortical gray-matter of Mindboggle (RRID: SCR 002438). Brain tissue segmentation of cerebrospinal fluid (CSF), white-matter (WM) and gray-matter (GM) was performed on the brain-extracted T1w using *fast* (FSL 5.0.9, RRID:SCR 002823).

For each of the functional runs, the following preprocessing was performed. First, a reference volume and its skull-stripped version were generated using a custom methodology of fMRIPrep. A deformation field to correct for susceptibility distortions was estimated based on a field map that was co-registered to the BOLD reference, using a custom workflow of fMRIPrep derived from D. Greve's *epidewarp.fsl* script (www.nmr.mgh.harvard.edu/~greve/fbirn/b0/epidewarp.fsl) and further improvements of Human Connectome Project Pipelines. Based on the estimated susceptibility distortion, an unwarped BOLD reference was calculated for a more accurate co-registration with the anatomical reference. The BOLD reference was then co-registered to the T1w reference using *bbregister* (FreeSurfer) which implements boundary-based registration. Co-registration was configured with nine degrees of freedom to account for distortions remaining in the BOLD reference. Head-motion parameters with respect to the BOLD reference (transformation matrices, and six corresponding rotation and translation parameters) were estimated before any spatiotemporal filtering using *mcflirt* (FSL 5.0.9). The BOLD time-series were resampled to surfaces on the following spaces: fsaverage and fsnative (FreeSurfer). The BOLD time-series were resampled onto their original, native space by applying a single, composite transform to correct for head-motion and susceptibility distortions. These resampled BOLD time-series will be referred to as "preprocessed BOLD". A reference volume and its skull-stripped version were generated using a custom methodology of fMRIPrep. A set of physiological regressors were extracted to allow for component-based noise correction[59]. Principal components are estimated after high-pass filtering the pre-processed BOLD time-series (using a discrete cosine filter with 128 s cut-off) for the anatomical CompCor (aCompCor). The time-series entering the CompCor analyses are derived from a mask at the intersection of subcortical regions with the union of CSF and WM masks calculated in T1w space, after their projection to the native space of each functional run (using the inverse BOLD-to-T1w transformation). Gridded (volumetric) resamplings were performed using *antsApplyTransforms* (ANTs), configured with Lanczos interpolation to minimize the smoothing effects of other kernels. Non-gridded (surface) resamplings were performed using *mri_vol2surf* (FreeSurfer).

To estimate voxel-wise responses to each orientation, a general linear model (GLM) was fit using SPM12 (version 7487, RRID:SCR 007037) to the time-series of each voxel during each orientation run. Fitting the GLM can be viewed as a preprocessing step to reduce the dimensionality of the data; the method presented here could be configured to run on the raw timeseries, but working with beta weights of a GLM rather than the raw timeseries drastically reduced the computational requirements of the Bayesian estimation. Prior to fitting the GLM, each voxel's timeseries was converted into a percent signal change, relative to the average signal within a run (across voxels). Design matrices were convolved with the canonical hemodynamic response function, parameterized with the SPM12 defaults, and additionally contained both six motion (three translation and three rotation) and multiple aCompCor regressors. For each run, the number of components was determined by the broken-stick method.

### Statistics and reproducibility

*Population receptive field mapping*. To mitigate the effects of stimulus vignetting—whereby orientation information differs between the edge and central portions of a grating[60,61]—we restricted the analyses to only striate voxels whose population receptive fields (pRFs) did not overlap with the stimulus edges. We estimated the pRFs of each voxel with standard methods. First, the preprocessed functional data for the pRF scans were converted into percent mean signal change within a run (across voxels). The compressive spatial summation model was fit to each voxel using analyzePRF HCP7TRET[52,57]. Following Benson et al., the compressive exponent of this model was set to 0.05. The resulting pRF parameters were combined with an anatomical prior for a Bayesian estimation of the parameters (neuropythy 0.94)[62]. Only the parameters of voxels for which the pRF explained more than 10% of the variance of the run were used as empirical parameters for Bayesian estimation; the remaining voxels' posterior pRF parameters were determined entirely by the prior.

The resulting pRF parameters determined whether a voxel would be retained for analyses. The pRF resembles an isotropic, bivariate Gaussian. The three pRF sessions were analyzed separately, resulting in three sets of three pRF parameters per voxel. Within a set of parameters, two indicate the center location of the pRF, and the third determines its size—the standard deviation of the Gaussian. A voxel was retained only if a circle centered on its pRF with radius equal to two standard deviations was entirely contained by the grating stimulus in each of the three sessions.

*Estimated average voxel tuning function*. The voxel tuning function (Fig. 4) was estimated according to a "binning" method[10]. First, each voxel's activity was z-scored, separately for each voxel. Then, one run was held out and the voxel's preferred orientation was calculated with the remaining runs. The preferred orientation was estimated as the orientation that produced the largest activity, averaged across runs and levels of contrast. Next, the activity in the held-out run was plotted against stimulus orientation translated into a function of distance from the voxel's preferred orientation. This procedure was repeated once for each available run, holding out a different run on each repetition. Averages and repeated-measures confidence intervals, across voxels and ignoring participants, were then calculated[63], treating contrast and distance from preferred orientation as within-voxel factors.

In Fig. 4, one session of one participant was excluded; the figure shows only sessions in which the participant was shown eight orientations.

*Bayesian estimation of parametric circular normal models*. Each of the voxel-specific parameters (baseline tuning function, tuning modulation, and variability in the output of the tuning function) were estimated with a hierarchical Bayesian model. All models were estimated with R (R Core Team) and its interface (RStan, 2.18.2) to the Stan language[64]. The Stan language provides a robust variation of Hamiltonian Monte Carlo[65,66], which efficiently and accurately approximates Bayesian posterior distributions. Two diagnostics were used to assess the accuracy of the estimation. First, we calculated the split-$\hat{R}$ for each parameter in the model[67]. This value is analogous to an F-score in an analysis of variance, and values close to 1 give no evidence that the different Markov chains are sampling from different distributions. The split-$\hat{R}$ was below 1.1 for all parameters[25]. Second, chains were monitored for divergences, whose presence would indicate that the samples, even if they are from a common distribution, likely misrepresent the true posterior distribution. There were no divergences when applying the multiplicative and additive models to real data.

In all applications of models to data, four chains were initialized with random values for the parameters (RStan defaults), and the sampling algorithm was given 1000 draws to adapt. After adaptation, each chain was used to draw 2000 samples from the posterior distribution. All model comparison was done with Pareto-smooth importance sampling, leave-one-out cross-validation plus, a technique for approximating cross-validation[28]. We refer readers to the original report for details of this technique.

*Model recovery*. For data-uninformed model recovery, 100 datasets were simulated from the priors of both models. The datasets contained one participant contributing 100 voxels measured in 18 runs at 8 levels of orientation. Both the additive and multiplicative models were applied to each simulated dataset, resulting in 200 simulated datasets and 400 model applications.

Data informed model recovery was implemented analogously to the data-uninformed model recovery. One hundred datasets were generated from the posterior of the multiplicative and additive models. The datasets contained six participants, each contributing 200 voxels measured in 18 runs at 8 levels of orientation. The multiplicative and additive models were applied to these simulated datasets and model comparison was conducted as above.

*Orthogonal regression estimate of slope*. A derivation of the slope for orthogonal regression can be found in Casella and Berger, chapter 12[27]. Here, we report the result. Let $x_i$ be the activity of a voxel at low contrast on trial $i$, out of $n$ trials; let $y_i$ be the activity of a voxel at high contrast on an analogous trial (i.e., the beta weight for the same orientation in the same run); let $\bar{x}$ be the voxel's average activity at low-contrast; and let $\bar{y}$ be the voxel's average activity at high-contrast. The following are the sums of squares and the cross-product.

$$S_{xx} = \sum_{i}^{n} \left( x_i - \bar{x} \right)^2$$

$$S_{yy} = \sum_{i}^{n} \left( y_i - \bar{y} \right)^2$$

$$S_{xy} = \sum_{i}^{n}(x_i - \bar{x})(y_i - \bar{y})$$

The slope, $g$, that solves the total least squares problem is given by

$$g = \frac{-\left(S_{xx} - S_{yy}\right) + \sqrt{\left(S_{xx} - S_{yy}\right)^2 + 4S_{xy}^2}}{2S_{xy}} \quad (5)$$

As stated in the main text, a separate slope was estimated for each voxel. In one session, one participant was shown seven rather than eight orientations. When calculating the slopes for this participant, the activity from those sessions were excluded.

*Orientation preference simulation.* To show that there are circumstances in which a von Mises density function is a reasonable weight distribution, we simulated cortical orientation preference maps (Fig. 2). Each voxel was constructed with a Kohonen network. We tailored the network parameters so that, following self-organization, each voxel would contain a roughly accurate number of pinwheels for a voxel of size 2.2 mm³—the voxel sized used in this study—approximately two pinwheels per millimeter[68]. However, this simulation is meant as a "proof of concept", rather than a demonstration that the von Mises distribution is an ideal weight distribution. That is, it does not consider many factors that likely influence how neuronal activity will be reflected in voxel activity[48]. For example, it does not consider the variability in pinwheel density across V1, vasculature, nor how cortical folding would allow disparate parts of cortex to be sampled by the same voxel. Nevertheless, it shows why we might expect weight distributions in most voxels to be relatively flat, with the deviations from uniformity being captured adequately by the single mode of the von Mises distribution.

Each network was simulated as follows. First, a grid of $67 \times 67$ "neurons" was initialized to have a random orientation preference. Then, for 20000 iterations, the grid was presented with a random orientation stimulus. The neuron that preferred an orientation closest to the stimulus was selected, and the weights were updated. Weights were updated based on the proximity of a neuron in the grid to the winning neuron. Specifically, on iteration $i + 1$, the weight, $w$, of neuron $j$ was given by the following:

$$w_{j,i+1} = w_{j,i} + \frac{\psi\left(d_j, \sigma_i\right)}{\psi\left(0, \sigma_i\right)} \delta_i \quad (6)$$

In Eq. 6, $\psi(x, \sigma)$ is the density of $x$ as assigned by the density function of a normal distribution with mean 0 and standard deviation $\sigma$. The variable $d_j$ is the distance on the grid between neuron $j$ and the winning neuron (Euclidean distance, as calculated after setting a city-block distance of 0.03 between adjacent neurons). The learning rate $\delta_i$ is the difference between the neuron's preferred orientation and the presented stimulus. Finally, $\sigma_i$ changes the effective distance between neurons over the course of learning, altering the size of the update to preferred orientation for neurons that neighbor the winning neuron. On each iteration, $\sigma_i$ is equal to $0.06 * ((20000 - i + 1)/20000)$.

**Reporting summary**. Further information on research design is available in the Nature Research Reporting Summary linked to this article.

## Data availability
The imaging and behavioral data that support the findings of this study are available as a repository on the Open Science Framework (https://doi.org/10.17605/OSF.IO/93YNM).

## Code availability
Software to perform the analyses is available on the Open Science Framework (https://doi.org/10.17605/OSF.IO/93YNM).

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

## Acknowledgements
The work was supported by NIH award 1RF1MH114277-01 to R.A.C. and D.E.H.

## Author contributions
Conceptualization: R.A.C. and D.E.H. Data curation: P.S. Formal analysis: P.S., R.A.C., and D.E.H. Funding acquisition: R.A.C. and D.E.H. Investigation: P.S., R.A.C., and D.E.H. Methodology: P.S., R.A.C., and D.E.H. Project administration: P.S., R.A.C., and D.E.H. Software: P.S., R.A.C., and D.E.H. Supervision: R.A.C. and D.E.H. Validation: P.S., R.A.C., and D.E.H. Visualization: P.S., R.A.C., and D.E.H. Writing—original draft: P.S., R.A.C., and D.E.H. Writing—review & editing: P.S., R.A.C., and D.E.H.

## Competing interests
The authors declare no competing interests
