## [Peer Review File · Communications Biology]

Reviewers' comments:

Reviewer #1 (Remarks to the Author):

A question of principal interest in visual neuroscience is how neurons encode properties of the environment, and how this encoding depends on changes in stimuli and/or task demands. At the single-unit level, different mechanisms have been observed, including additive shifts in tuning profiles, multiplicative gain, changes in tuning bandwidth, and combinations of all of these. At the fMRI level, where individual fMRI voxels assay hemodynamic signals which aggregate over hundreds of thousands of neurons, it has been challenging to discriminate between these mechanisms due to the intractable inverse problem (hundreds of thousands of neurons to a voxel). In this manuscript, the authors describe a modeling procedure that attempts to identify which mechanism best accounts for changes in voxel response properties measured across multiple experimental conditions (here, high and low contrast) and feature values (here, grating orientations). They conclude that, even though looking at aggregate data suggests an additive shift mechanism, the data can be better described by a multiplicative gain model. They also describe a model-free method for testing between different encoding mechanisms.

I think the authors are pursuing an interesting question, but the manuscript reads as incomplete in its present form. The modeling procedures are poorly described, the results are only minimally presented, and some of the modeling assumptions are not well-motivated. Accordingly, this manuscript would require extensive revisions before it could be considered acceptable for publication. I describe these concerns below.

Data presentation: At present, I only see two data figures (Fig. 3 and 5), one of which shows 'voxel tuning functions' measured across experimental conditions, and the other shows some aggregate results of the non-parametric analysis approach described. For a manuscript describing a new modeling procedure, I think it's critical for the authors to clearly and completely detail the data analysis approach at all stages. For example, a figure showing raw data associated with the 'slope' analysis for many example voxels would be very useful. Additionally, the Results section is extremely sparse – I expected some direct comparison of different modeling/analysis choices, a walkthrough of how the steps of the analysis can be visualized with actual data (as mentioned above), and, ideally, an example of a clear visualization of the efficacy/utility of the modeling approach. At the very least, examples of comparisons between the model fits (additive/multiplicative) for a few example voxel would be useful for readers.

"Neuromodulation modeling": While I understand the authors need to provide their technique with a memorable, descriptive, and useful name, I'm not sure 'neuromodulation modeling' is an appropriate choice. To most neuroscientists, neuromodulation refers to the actions of neuromodulators on (electro)physiological properties of cells. Attention and contrast manipulation may invoke these systems, but the implication of this nomenclature if the technique measures the results of neuromodulators – but instead it just is meant to measure changes in neural tuning.

Assumptions regarding clustered tuning: it seems that a key assumption of the parametric modeling framework is that the distribution of feature preferences within a voxel follows a circular Gaussian. The authors attempt to demonstrate the validity of this assumption in Fig. 2, but I don't find these data (derived from simulations) to be very compelling. The histograms don't appear to be very (circular) Gaussian to me. To strengthen their report, the authors should demonstrate the extent to which their technique is effective when this assumption is not met. (another assumption that seems strange: lines 95-96 stating that neurons within, but not across, voxels have same TF shape)

Conceptual level: The point of this method is to build an analysis framework whereby one could infer the most likely change in single-unit encoding models that best accounts for the observed response changes across a population of voxels, which each span a subpopulation of neurons (including both tuned and untuned cells). The authors need to be more clear that there remains a fully intractable inverse problem that, fundamentally, cannot be solved. This method seems to help clarify the relative likelihood of different neural encoding changes, but it will always remain possible that the tuning properties of different neurons in a voxel undergo a complicated and non-uniform combination of changes. Could this method ever reveal such a result? And if not, what does it mean to infer the most likely mechanism based on data measured at the voxel level, when the reality is that the brain co-opts a variety of mechanisms to support its dynamic processing demands? This is a critical point for discussion.

Model prediction accuracy: within a condition (e.g., low contrast), how well does the best-fit model

(fit, e.g., to all fMRI runs except one) predict responses to the held-out run? This is an ideal first step to testing any fMRI encoding model (see, among many others, Liu et al, 2018, J Neuro, cited by the authors, for a clear argument about why this is important).

Ground truth simulated data: I see that the authors briefly describe some data-uninformed model recovery procedures, and that these resulted in accurate model identification. Can the authors show this data? Does the model recovery procedure produce qualitatively-consistent voxel tuning functions (like plotted in Fig. 3) compared with the observed fMRI data? How accurately does the hierarchical model recover the ground-truth parameters of the generating dataset? Does this depend on noise, number of voxels, etc? And what if data is generated using a mechanism distinct from that built in to the modeling (below), or a combination of mechanisms (above)?

Additional mechanisms: the report, at present, only focuses on baseline/additive shifts and multiplicative gain. What about changes in tuning width, tuning concentration, or tuning preference shifts? These are all also observed in different scenarios that could be of interest, and, should they be occurring, could mask themselves as one or another of the mechanisms considered in the report. I see the authors' argument in lines 134-140, but this only applies to the present dataset, and would not apply to simulated 'ground truth' data.

Nonparameteric analysis: the authors mention that the slopes can take on nearly any value when the difference in signal is small using their orthogonal regression method. Couldn't this be accounted for by resampling or bootstrapping the regression across scanning runs? Also, the authors should clarify that this procedure is only effective for the linear mechanisms considered here (additive shift/multiplicative gain). I'm not sure this would work effectively for the other types of tuning mechanisms mentioned earlier in the manuscript and above in my review.

Additional minor comments:

Lines 22-23: Fig. 1A shows slices through a 2D tuning profile (SF x orientation) – is this really analogous to changes in tuning as a function of cognitive state? Or is this just a more complete characterization of the (non)linear filter of the cell?

Lines 29-30: this introduces the idea of a 'voxel tuning function' without defining/describing how one might be measured

Fig. 1 caption: usually change in fluorescence is listed as uppercase $\Delta F/F$

Lines 98-99: low-contrast is assumed to be a 'baseline' condition – wouldn't this be a lower-SNR scenario, so the model fits would be noisier? I'd think you'd want to use the highest-SNR data possible for model fitting. Perhaps the Bayesian modeling accounts for this somehow – regardless, the authors should add text convincing readers this isn't something to be concerned about

Fig. 3: what ROI(s) is this? Number of subj? CI's across...subj? voxels?

Line 131: reference to missing Figure C1

Eq 4a: the parameter list prior to the = contains a parameter alpha, but the term to the right of = contains only a? The text in the preceding paragraph is also confusing about what the alpha and a parameters actually are – the authors should clarify.

Fig. 5A: is this plotting hierarchical model fits? Or data? Or both? This, I think, is the closest the authors get to plotting real data for evaluating their modeling procedure, but it's very hard to understand what's going on in the figure at present

Lines 265-266: "unreflective of the response properties of the underlying, individual neurons" – this doesn't seem to be an accurate characterization of VTFs. Maybe changes in encoding properties measured with tools like fMRI don't unambiguously demonstrate a particular coding mechanism, but I think it's going a bit too far to say the tools give results that don't reflect the underlying neural response properties at all, which is what's implied at present

Lines 282-293: this paragraph of the discussion implies that SPM methods are more 'fine-grained' than multivoxel methods – I don't understand the authors' argument here; these are extremely different approaches that can't easily be compared this way

Lines 306-307 (and this paragraph) attempt to distinguish the current approach from other encoding model methods, and point out that the choice of a baseline condition in encoding model techniques is arbitrary. However, isn't that the case here too? What's less arbitrary about picking a low-contrast baseline here as compared to the reports cited?

Lines 321-323: I don't understand this sentence in context. Doesn't the NMM framework involve an explicit form of the neural (and voxel) tuning functions?

Line 369: I don't think Fig. 3 looking unimodal can be used in any way to support the use of a unimodal weight function for orientation preference within each voxel. It's likely there are a very large number of non-unimodal voxels that contribute. (for example, generating random mean

responses for each voxel, then generating a synthetic data with noise; sorting voxels according to the max of a held-out dataset, and plotting the average response from the remaining data – the standard VTF approach applied to noisy synthetic data with no unimodal voxel tuning function – is likely to result in a unimodal VTF).

Line 443: what was/were the inter-trial intervals?

pRF methods: what was the area of the screen mapped? What was/were the size(s) of the bars/wedges?

ROIs: how were these defined? (I assume using pRF parameters) and which ROI(s) were used throughout? I think I saw one reference to primary visual cortex, but this should be made extremely clear

Lines 545-546: why were the 3 pRF sessions analyzed independently? And how were the parameters combined across sessions? (or, instead, were the within-session parameters used to select voxels?)

Lines 544-550: use of 2 SD-radius circle seems overly conservative. How many voxels were selected per participant/ROI?

In the Results, the authors mention an “approximation to cross-validation” in the methods, but I couldn’t find any specific mention of this in the Methods.

Reviewer #2 (Remarks to the Author):

1. This review is tricky because I think the basic aim of this paper (forward modelling of fMRI data) is really important, and the general approach taken to this problem is excellent and sophisticated, but the example application to the simple case of distinguishing multiplicative vs additive models of visual contrast is a somewhat trivial, and the proof for the “non-parametric check” in the supplementary material undermines the rationale for the parametric model (in showing that multiplicative a versus additive modulation can be distinguished simply by slope of activation plots, without fitting data). Is there another empirical test to which the parametric model could be applied, to demonstrate the need for parametric model fitting, i.e. that cannot be shown a priori to be testable with a simpler data analysis technique? Or if I am missing the point, perhaps the authors could explain why the parametric model is so important for the empirical case they chose?

2. I could not find code for the models on the original OSF link given, but thank the authors (and editor) for providing the link in a separate email – I hope this link will be included in the final version of the paper. Given that my review was already overdue because of the delay entailed by me asking for the code, I did not have time to run stan and the associated functions, but I did look through them, and it helped better understand the models. If there is a revised submission (and more time to update the code repository as the email suggested), I might have time to test the code then. One minor observation: I think there may be a literal-code-and-paste error in the README.md of <https://gitlab.com/psadil/vtuner/>, where the code below the section for the “Additive” model looks identical to that for the “Multiplicative” model above?

3. Line 95: what if one allows neural tuning functions to vary in shape within a voxel? This seems possible in reality – so if this assumption is dropped, can any inferences still be made, or does the whole framework become inestimable (at least without strong priors on the variation in shapes within a voxel)?

4. In the paragraph beginning on line 134, is it possible that the K_v parameter does change with contrast, and it *could* change the average activity of a voxel, if one dropped the unit-normalisation of the integral of the von Mises function(f)? It seems likely to me that some neurons with a preference close to the stimulus orientation could change their concentration as contrast changes, while other neurons with preferences further away do not change at all?

5. On line 384, it is stated “...applying NMM to explore neuromodulations beyond multiplicative gain and additive shift, further model recovery simulations would be required.” It would really strengthen the paper (though I suspect the authors will groan, and argue this is future work!) if they explored model recovery for a much larger range of variables in the models (eg Point 4 above). I suspect that the spatial limitations of fMRI will mean that the “inverse” problem of

inferring neural properties from voxel-level data becomes too ill-posed, such that useful inferences are only possible for a very small subset of possible hypothetical neural effects (like multiplicative vs additive effects). Perhaps I am wrong, but I think this would be a very influential paper: i.e., documenting what type of neural hypotheses can be distinguished by fMRI and what types can NOT (once the complexity of the mapping is modelled, as the authors do here). This would back-up the authors' claim on line 402 that "Despite the simplicity of the test case, the method should be applicable to a broad range of domains and manipulations."

Minor points

6. In the Discussion, it is stated that "Like biophysical models, NMM connects the latent neural activity to the observed BOLD, by assuming a linear coupling that is justified in our experimental design..." In most current biophysical models, the BOLD response to neural activity has several nonlinearities, even if a linear convolution model is often assumed in fMRI analysis (so I don't think the "Like biophysical models" bit is correct). This linear approximation may be sufficient for the temporal profile of responses to successive brief stimulations (used in most fMRI designs), at least if the time between brief neural bursts (as determined by SOA) is long enough, but the BOLD response has been claimed to be a nonlinear function of other variables like the duration of neural activity and possibly stimulus contrast (see Vazquez & Noll, 1998, NeuroImage). Even putting to one side debates about whether nonlinearities as a function of contrast are neural or haemodynamic in origin, I'm not sure many other types of nonlinearity matter for the present model, which is a spatial rather than temporal model – i.e., has no concept of duration of neural activity, so could not be applied to designs with different stimulus durations (without extension to the temporal domain). In short, while it's good to acknowledge that the current model assumes a linear neural-to-BOLD mapping, it might help for the reader to consider precisely what types of nonlinearities (i.e. as function of what types of experimental variables) might be relevant to the model's predictions.

7. Line 193: isn't "orthogonal regression" the same as what is commonly known as "Deming regression"? This doesn't necessarily need to be changed; just might be better known to some by this name (cf von Mises distributions!).

8. Line 234, for those less familiar with Bayesian model comparison, could some intuition be given for the "expected log pointwise predictive density", eg guidelines for how impressive 14.5 standard errors is?

Reviewer #3 (Remarks to the Author):

The authors have proposed an interesting and novel model to link a voxel's BOLD response to neural activity. As they have correctly mentioned, their model can have applications in investigating the effect of cognitive states on perception. However, their model is based on some simplifying assumptions, and it is not clear how they will impact the results. One of these assumptions is the homogeneity of the changes of the neural responses, which is far from reality. Even in the references, they have provided for their "ground truth" case, the observations have suggested that not all the neurons show a boost in orientation tuning function with contrast, and there are neurons that reach saturation at very low contrasts, below the 50% contrast level used in this study; the point that has not been even mentioned after citing these studies. They have admitted to this limitation in their discussion but have neither discussed the possible misjudgments that may arise due to this simplifying assumption nor the way the non-linearity of the neural responses can be incorporated in the model. They argue that even with this simplifying assumption the NMM has been able to detect the correct model, but picking the right model from two options and in a simple case cannot be a strong proof for the validity of the model in more complicated situations and especially when it is supposed to be used for discovering neural dynamics in novel conditions. Moreover, this limitation has not been touched upon in the abstract. The given image of the NMM in the abstract as a complete model that can be used in all conditions and without limitations is somehow misleading.

There is also some information about the fMRI analysis that seems to be missing from the manuscript. It has not been stated clearly from which area the voxels have been selected. Is figure 3 representative of voxels of the V1 area or V1-V3? And how many voxels were left after applying a threshold in figure 5?

I also suspect that the additive mechanism seen in figure 3 arises due to including all the voxels and excluding the less active voxels with the same thresholding approach may change this figure. Have the authors tried regenerating this figure with only the most active voxels?

Minors:

- The purpose of the proposed tool is to link voxel's response to neural response and it seems to me that the NMM name that only focuses on neuromodulation does not communicate this purpose.

- Von Mises has been used before for model fitting on the orientation tuning curves. I suggest citation of Swindale 1998.

- Title of figure 2 does not convey the reason behind the analysis. The analysis has been done to demonstrate that von Mises weight function is a reasonable model, but the title suggests that it is only a simulation of orientation preferences and the text does not clarify the purpose either. It did not become clear to me until I reached the methods section.

- Line 131, Figure C1 does not exist.

- Line 147, how the "weakly informative" priors have been selected? Have the parameters been set based on the literature?

- Line 171, the shape of the neural tuning function and the weight distribution do not seem to be totally irrelevant. According to the given proof in the supplementary methods, $q(r)$ function, which is the convolution of these two, should be an invertible function for the non-parametric check to be true.

- Figure 5A: Contrary to the caption content, some slopes seems to be larger than 1 in the additive model figure. Adding a histogram of the slopes for the additive model as well may help resolve this confusion.

- Supplementary material, equation 6, wrong sign: $-ga$

Reference:

Swindale, Nicholas V. "Orientation tuning curves: empirical description and estimation of parameters." *Biological cybernetics* 78.1 (1998): 45-56.

Reviewer #1 (Remarks to the Author):

A question of principal interest in visual neuroscience is how neurons encode properties of the environment, and how this encoding depends on changes in stimuli and/or task demands. At the single-unit level, different mechanisms have been observed, including additive shifts in tuning profiles, multiplicative gain, changes in tuning bandwidth, and combinations of all of these. At the fMRI level, where individual fMRI voxels assay hemodynamic signals which aggregate over hundreds of thousands of neurons, it has been challenging to discriminate between these mechanisms due to the intractable inverse problem (hundreds of thousands of neurons to a voxel). In this manuscript, the authors describe a modeling procedure that attempts to identify which mechanism best accounts for changes in voxel response properties measured across multiple experimental conditions (here, high and low contrast) and feature values (here, grating orientations). They conclude that, even though looking at aggregate data suggests an additive shift mechanism, the data can be better described by a multiplicative gain model. They also describe a model-free method for testing between different encoding mechanisms.

I think the authors are pursuing an interesting question, but the manuscript reads as incomplete in its present form. The modeling procedures are poorly described, the results are only minimally presented, and some of the modeling assumptions are not well-motivated. Accordingly, this manuscript would require extensive revisions before it could be considered acceptable for publication. I describe these concerns below.

Data presentation: At present, I only see two data figures (Fig. 3 and 5), one of which shows ‘voxel tuning functions’ measured across experimental conditions, and the other shows some aggregate results of the non-parametric analysis approach described. For a manuscript describing a new modeling procedure, I think it’s critical for the authors to clearly and completely detail the data analysis approach at all stages. For example, a figure showing raw data associated with the ‘slope’ analysis for many example voxels would be very useful. Additionally, the Results section is extremely sparse – I expected some direct comparison of different modeling/analysis choices, a walkthrough of how the steps of the analysis can be visualized with actual data (as mentioned above), and, ideally, an example of a clear visualization of the efficacy/utility of the modeling approach. At the very least, examples of comparisons between the model fits (additive/multiplicative) for a few example voxel would be useful for readers.

We thank you for pointing out ways in which the modeling procedure can be clarified. The manuscript has been thoroughly reworked in response and here we outline a few specific changes:

- Regarding the data reported in the previous Figure 3 (now Figure 4), we now provide in the results section an expanded treatment of the “binning” method that was used to create this figure. The binning method is a means of revealing tuning of responses in the aggregate across voxels. It involves using held-out data to sort voxels into “bins” of orientation preference, then plotting their responses across orientation using the remaining data. An average voxel response is then plotted by circularly shifting all voxels such that the response functions are aligned, using preferred orientation as the zero point.
- In response to the request for “a figure showing raw data associated with the ‘slope’ analysis for many example voxels”, we have updated Figure 5 to show the individual slopes for all voxels derived from applying orthogonal regression to the empirical data (5A, left panel).

- In response to the request for “a clear visualization of the efficacy/utility of the modeling approach” we now provide a visualization of how well the Bayesian-estimated circular Normal model reproduces the empirical fMRI data (4B with model posterior prediction curves passing through BOLD data).
- In response to the request for “examples of comparisons between the model fits (additive/multiplicative) for a few example voxels”, we augmented Figure 4 (previously Figure 3) by showing the posterior predictions of the two models plotted against every observed datapoint (4C). The models make superficially very similar predictions and the superiority of the Multiplicative model in model selection (i.e., the fact that it provides a superior account of the modulation of neural level tuning) derives from its ability to capture *subtle* features of the empirical data that are not easily visualized in a plot of aggregate-level data. Instead, the superiority of the multiplicative model is built up from many small improvements across all voxels.

Below, these updated figures are reproduced.

Figure 1 Average voxel tuning functions visually suggest an additive shift even though the multiplicative model is better in its posterior predictions of individual voxel data. **A)** Average voxel tuning at low and high contrast using binning to average (see Methods). Error bars give 95% confidence interval for the mean of the 1,010 V1 voxels across seven participants. **B)** Examples of individual voxel BOLD data in comparison to posterior predictions of the multiplicative gain model. The panels show the 16 voxels with the largest average difference in activity across levels of contrast. Vertical error bars span 95% confidence intervals (within-subjects), and the width of the lines indicate the 95% highest density interval of the posterior predictive distribution for the multiplicative model (additive model looks similar). **C)** Overall comparison between each model ($\hat{\beta}$) and observed data (β). Each point corresponds to a single voxel's average activity at a given orientation at a given level of contrast. Panels show the average of the posterior for predictions from either the additive or multiplicative model.

*Figure 2 Non-parametric check of model comparison using orthogonal regression analysis. A) Orthogonal regression on data and model predictions. Each line corresponds to a single voxel. Most voxels are weakly responsive at each level of contrast and are only weakly influenced by the experimental manipulation (i.e., lines cluster in the lower left corner, close to the diagonal). Across voxels (i.e., comparing separate lines), both models capture larger visual contrast effects for more responsive voxels: predictions lying further rightward along the x-axis are located further away from the diagonal. However, only the multiplicative model captures the observed interaction between orientation and contrast within voxels. Compare to **Error! Reference source not found.** B) Distribution of within-voxel slopes. As allowed by multiplicative but not additive modulation, the distribution of slopes is shifted above 45° (i.e., slope of 1).*

“Neuromodulation modeling”: While I understand the authors need to provide their technique with a memorable, descriptive, and useful name, I’m not sure ‘neuromodulation modeling’ is an appropriate choice. To most neuroscientists, neuromodulation refers to the actions of neuromodulators on (electro)physiological properties of cells. Attention and contrast manipulation may invoke these systems, but the implication of this nomenclature if the technique measures the results of neuromodulators – but instead it just is meant to measure changes in neural tuning.

This is an excellent point. In response, we have changed the name of the framework to “Inferring Neural Tuning Modulation” (INTM) and the manuscript no longer makes any reference to neuromodulation. The title of the manuscript was also changed to reflect this new terminology.

Assumptions regarding clustered tuning: it seems that a key assumption of the parametric modeling framework is that the distribution of feature preferences within a voxel follows a circular Gaussian. The authors attempt to demonstrate the validity of this assumption in Fig. 2, but I don’t find these data (derived from simulations) to be very compelling. The histograms don’t appear to be very (circular) Gaussian to me. To strengthen their report, the authors should demonstrate the extent to which their technique is effective when this assumption is not met. (another assumption that seems strange: lines 95-96 stating that neurons within, but not across, voxels have same TF shape)

We agree that the assumption of a unimodal distribution of feature preferences within a voxel is likely incorrect for some voxels and potentially problematic for the Bayesian-estimated circular Normal model. Providing some reassurance, we have added additional plots of the data (for 16 individual

voxels in Figure 4A), showing that the individual voxel-wise tuning functions appear well characterized by unimodality. However, a unimodal voxel tuning function can arise from a multimodal distribution of feature preferences in some cases. It is precisely for this reason that we developed the slope test, which does not rely on this assumption (neither does the slope test rely on the assumption that neurons within a voxel have the same tuning shape). By analogy, many parametric statistical tests assume normally distributed errors, which is often violated. In such cases, statistical conclusions are strengthened if less statistically powerful non-parametric tests (i.e., ones that don't make this assumption) indicate the same result. In using INTM, we recommend use of the non-parametric slope test as an additional check on the results without making these assumptions. The primary assumption of the slope test is that all feature-tuned neurons within a voxel have the same level of additive change and the same level of multiplicative change; no assumptions are necessary regarding the shape of the neural tuning functions or the shape of the distribution of feature preferences.

The distinction between the parametric model, with assumed circular Normal functions, and the non-parametric slope test is now made throughout the manuscript, including the abstract.

At the start of the section introducing the non-parametric slope test (Lines 234 - 255), we now write

“The parametric model assumed normal functions for the NTFs and weight distributions and additionally assumed homogeneity of NTF shape within a voxel. These assumptions will often be violated, but such violations might not pose a problem for model comparison, particularly if the violations of these assumptions are uniformly applied across the stimulus dimension of interest. Nevertheless, as a complement to parametric model comparison, we developed a more qualitative non-parametric check of the empirical data. If this check supports the same conclusion as the parametric model, this suggests that any violations of parametric assumptions were insufficient to alter the theoretical conclusions.

The parametric model is crucial to the development of the non-parametric check and the specific parametric models under consideration should first be used to generate predictions for the check. The consequences of parametric assumptions are then considered in light of these predictions. In the case of a comparison between multiplicative gain versus additive shift, these two forms of tuning modulation can be distinguished using voxel-wise regression; if the modulation is multiplicative, a within-voxel plot comparing high- versus low-contrast should have a slope greater than 1, whereas if the tuning modulation is additive, this plot should reveal a slope equal to 1. Furthermore, these predictions do not rely on the parametric assumptions. For instance, if a voxel is multimodal, preferring more than one orientation, this would serve only to rearrange the order of the orientations in the regression plot, but the qualitative distinction between a slope of 1 versus a slope greater than 1 would still map onto additive shift versus multiplicative gain. The generality of the slope test across all shapes of tuning function and all shapes of weight distribution is mathematically proved in Supplementary Methods Equations 1-4.”

This mathematical proof indicates that different voxels can take on different tuning shapes and yet the slope test will still differentiate between the models. We preview this at the end of the section introducing the parametric model, writing (Line 181 - 192):

“The parametric model assumes that all neurons within a voxel share a common shape for their tuning function (homogeneity of tuning shape), which is unlikely to hold strictly true. However, some degree of tuning shape heterogeneity should not invalidate inferences drawn about the form of tuning modulation, provided that tuning function shape does not differ systematically as a function of the orientation preference of the neurons. That is, there could exist a range of tuning function shapes, but if that heterogeneity occurs to the same extent for neurons centered on all preferred orientations, then it should not lead to scenarios where one form of modulation mimics another at the voxel level. Nevertheless, to assess whether such heterogeneity posed a problem when adjudicating between the forms of tuning modulation, we also developed a non-parametric check that does not make this assumption. If the non-parametric check supports the same qualitative conclusion as the parametric model, this suggests that the homogeneity of tuning shape assumption was adequate.”

The proof provided in Equations 1-4 in the Supplementary Methods is different than the proof provided with our initial submission. This revised proof is much more general, demonstrating that the non-parametric slope test doesn't even assume that all neurons within a voxel have the same tuning shape.

Conceptual level: The point of this method is to build an analysis framework whereby one could infer the most likely change in single-unit encoding models that best accounts for the observed response changes across a population of voxels, which each span a subpopulation of neurons (including both tuned and untuned cells). The authors need to be more clear that there remains a fully intractable inverse problem that, fundamentally, cannot be solved. This method seems to help clarify the relative likelihood of different neural encoding changes, but it will always remain possible that the tuning properties of different neurons in a voxel undergo a complicated and non-uniform combination of changes. Could this method ever reveal such a result? And if not, what does it mean to infer the most likely mechanism based on data measured at the voxel level, when the reality is that the brain co-opts a variety of mechanisms to support its dynamic processing demands? This is a critical point for discussion.

This is a good point and we apologize for not being more upfront in our initial submission. We have mentioned this throughout the manuscript, including the abstract, which now says (Lines 5 - 8):

“Precise specification of neural tuning from the BOLD signal is not possible. Instead, INTM compares theoretical alternatives for the form of neural tuning modulation that might underlie changes in BOLD across experimental conditions.”

The third paragraph of the introduction now says (Lines 41 - 43):

“Rather than quantitatively solving this inverse problem, which is probably not possible, we developed a framework that allows drawing qualitative conclusions about changes in neural level tuning from the Blood Oxygen Level Dependent (BOLD) signal.”

Finally, the third to last paragraphs of the discussion now say (Lines 469 - 487):

“Despite its generality, INTM has two limitations that must be plainly acknowledged. The first is that INTM requires that the models under consideration make distinct predictions at the level of the voxel. For instance, the parametric and non-parametric INTM techniques both assume that the magnitude of tuning modulation is the same for all neurons that contribute to the voxel response. These techniques may be robust to relaxing this assumption if modulation magnitude difference apply uniformly across the weight distributions that maps neural preferences into voxel preferences. But if the magnitude of modulation varies systematically, there might no longer be distinct predictions at the level of the voxel. For example, if the magnitude of the additive shift is larger for neurons that prefer the same orientation as the voxel, this unlikely circumstance could produce a voxel tuning function that is indistinguishable from a constant magnitude of multiplicative gain. The inability to discriminate these two scenarios exemplifies the fundamental inverse problem that INTM cannot solve: INTM cannot determine the parameters of tuning functions for individual neurons (indeed, it is not designed to). Instead, INTM operates with distributions of neurons (grouped by, e.g., region-of-interest, voxel, feature preferences), and so it can only compare models whose distributions of tuning functions change in ways that are distinct. With the example of a magnitude of additive shift that happened to covary with the weight distribution, this might occur for some voxels, but is unlikely to occur with most voxels, demonstrating the need to use all of the voxels in the model comparison.”

Model prediction accuracy: within a condition (e.g., low contrast), how well does the best-fit model (fit, e.g., to all fMRI runs except one) predict responses to the held-out run? This is an ideal first step to testing any fMRI encoding model (see, among many others, Liu et al, 2018, J Neuro, cited by the authors, for a clear argument about why this is important).

We agree that cross-validation is a powerful method for assessing the model. In our initial submission, we failed to highlight that we used cross-validation when deciding which model provided a better account of the data. Cross-validation was performed using the PSIS-LOO technique, where LOO stands for "Leave One Out". In our initial submission, we also neglected to include a "posterior predictive check", which is a comparison between the model predictions after being applied to the data and the actual observed data. This is an important step in Bayesian estimation, demonstrating that the model can account for the data. I.

Ground truth simulated data: I see that the authors briefly describe some data-uninformed model recovery procedures, and that these resulted in accurate model identification. Can the authors show this data? Does the model recovery procedure produce qualitatively-consistent voxel tuning functions (like plotted in Fig. 3) compared with the observed fMRI data? How accurately does the hierarchical model recover the ground-truth parameters of the generating dataset? Does this depend on noise, number of voxels, etc? And what if data is generated using a mechanism distinct from that built in to the modeling (below), or a combination of mechanisms (above)?

We thank you for pointing out ways in which we can enhance the presentation of the model recovery simulations. We have addressed these questions in the following ways

- As stated above, Figure 4C now plots the posterior predictions of the two models, which shows a close correspondence between observed data and data generated from each model.

- The lines in Figure 4B show the average prediction of the multiplicative model for individual voxels as compared to the data, and Supplementary Figure 5 also plots predictions from the multiplicative model but in this case the plots include predicted variability (which is a critical aspect of model recovery).
- Supplementary Figure 6 shows parameter recovery for the population-level parameters in the data-informed model recovery.

Supplementary Figure 6 Parameter Recovery for Data-Informed Model Recovery. Distributions show the difference from true value across draws when (A) the additive model was fit to datasets generated from the additive posterior predictive distribution, or (B) the multiplicative model was fit to datasets generated from the multiplicative posterior predictive distribution. In all cases, the distributions are centered on 0, indicating lack of bias in the estimated parameters. Panels give parameters (compare to Supplementary Figure 1): α_loc : μ^α ; α_scale : σ^α ; γ_loc : μ^γ ; γ_scale : σ^γ ; κ_loc : μ^κ ; κ_scale : σ^κ ; $ntfp_loc$: either μ^α or μ^g ; $ntfp_scale$: either σ^α or σ^g ; σ : μ^σ .

Rather than providing more detail about data-uninformed recovery, we have instead provided more detail about data-informed recovery. We have clarified this in our presentation of data-informed recovery Lines (213 - 232):

“There is no guarantee that a specific real dataset will be diagnostic regarding a particular model comparison. For instance, if none of the voxels are sufficiently well-tuned, then the two forms of tuning modulation will make the same prediction. Similarly, if the manipulation of visual contrast is too weak, it would not be possible to reach a reliable conclusion. Whether the real dataset is sufficient for differentiating between the candidate models is assessed using data-informed model recovery by repeating the recovery process using what has been learned about the parameters of the models from the empirical data.

Whereas data-uninformed model recovery assesses the fairness of model comparison across a wide range of individual parameters that are plausible given the prior distribution, data-informed recovery generates synthetic data by sampling from the posterior distribution obtained by estimating the models with empirical data. Generating synthetic data that are consistent with real data constrains the parameters (e.g., tuning width and modulation) to a much narrower and more realistic range of values. Indeed, the behavior from the posterior parameter values closely matches the weak tuning effects seen in the real data at the level of separate voxels (e.g., Supplementary Figure 5). Because the data-uninformed model recovery has priors that are only weakly informative, it can allow extreme parameter values for estimated beta-weights, and hence implausibly sharp tuning functions; this increases the chance that each generating model will produce a unique signature pattern in the data, making model recovery likely to succeed. By contrast, the more constrained data-informed model recovery presents more of a challenge to model recovery, providing greater reassurance in the case that it succeeds.

Regarding the possibility of more complicated combinations of modulations, the second to last paragraphs of the Discussion now says (Lines 488 -501):

“The second key limitation is that INTM can differentiate only between models that are formally included in the model comparison process. If the “true” form of tuning modulation is not included in the set of possibilities, the results may mislead researchers into conflating the winning form of modulation with the true form. For instance, in the present study, the technique contrasted tuning modulation models in which just one form of modulation occurred but did not consider more complex situations in which multiple forms of tuning modulation could have occurred. For example, it is known from electrophysiology that with extended adaptation, orientation tuning functions can both widen and shift their preferred orientation⁵³. Since that more complex model, in which tuning widened and shifted within every trial, was not included in model comparison, INTM is silent about whether that effect occurred. Note that such combinations of tuning modulation could be included in the model comparison process, but when attempting to adjudicate between complex models it would be critical to use model recovery simulations, specifically data-informed recovery, to determine if the collected data were sufficiently constraining.”

Additional mechanisms: the report, at present, only focuses on baseline/additive shifts and multiplicative gain. What about changes in tuning width, tuning concentration, or tuning preference shifts? These are all also observed in different scenarios that could be of interest, and, should they be occurring, could mask themselves as one or another of the mechanisms considered in the report. I see the authors’ argument in lines 134-140, but this only applies to the present dataset, and would not apply to simulated ‘ground truth’ data.

This is an important question and in our initial submission we failed to point out that we considered other forms of tuning modulation and we also failed to provide the reader with some description of how to extend the framework to other forms of modulation. We now make this clear in the presentation of the parametric model writing (Lines 138 - 146):

“Within the INTM framework, each form of tuning modulation is instantiated by incorporating additional parameters that modulate one or more parameters in the baseline voxel tuning function (Equation 3, Figure 1C-D). In this validation test case as applied to visual contrast manipulations, we considered two forms of tuning modulation that are capable of producing an average increase in neural activity: multiplicative gain and additive shift. Other forms of neural tuning modulation, such as bandwidth changes and tuning preference shifts, could be implemented within the INTM framework, but these forms were not included in the candidate set of models for this test case because they do not change average neural activity (i.e., they are not viable models for the neural modulation that underlies changes in visual contrast).”

Later on in this section of the paper, we highlight how an end-user of the framework could address these other kinds of modulation (Lines 168 - 173):

“For other applications, shifts in orientation preferences could be modeled by allowing φ_v to change across conditions, and a change in concentration could be modeled by allowing κ_v to change across conditions. In the current dataset, these forms were not pursued because neither allows contrast to alter the average activity of a voxel across orientations – an effect that was clearly present in the empirical data (Figure 4A) – and so these forms could be rejected without explicit modeling.”

We have explicitly avoided making too many claims regarding which forms of tuning modulation will be identifiable. That is because identifiability depends on factors that will be particular to each experiment – like the manipulation of interest, stimulus feature, study population, etc. To reiterate the end of the penultimate paragraph of the Discussion (Lines 499 - 501) “...when attempting to adjudicate between complex models it would be critical to use model recovery simulations, specifically data-informed recovery, to determine if the collected data were sufficiently constraining.”

Nonparametric analysis: the authors mention that the slopes can take on nearly any value when the difference in signal is small using their orthogonal regression method. Couldn't this be accounted for by resampling or bootstrapping the regression across scanning runs? Also, the authors should clarify that this procedure is only effective for the linear mechanisms considered here (additive shift/multiplicative gain). I'm not sure this would work effectively for the other types of tuning mechanisms mentioned earlier in the manuscript and above in my review.

Regarding reliability of the slope estimate, the Bayesian version of the non-parametric analysis, reported in the *Supplementary Information*, stabilizes the estimates by modeling them hierarchically. In our previous analysis, the slope values from "untuned" voxels were particularly problematic because the best fit might be a nearly vertical line, corresponding to a slope of infinity (or negative infinity), which is a problematic measurement scale. In this revision we have remedied this measurement scale issue by using angles rather than slopes. Unlike the slope, the angle is bounded, and it is equally sensitive to changes in squared Euclidean distance within the bounds (that is, the angle of the line changes proportionately to the squared Euclidean distance of the points from the line at all possible angles; in contrast, the *slope* of the line changes far more for a given change in squared Euclidean distance when the line is near vertical than when it is near horizontal).

We describe this in the manuscript as follows (Lines 287 - 292):

“Finally, even a within-voxel slope analysis may be problematic because most voxels are only weakly tuned, responding to all orientations almost equivalently. For those voxels, the data for the slope analysis will be an uncorrelated cloud of points, and an orthogonal regression for a cloud of points can take on nearly any slope value, including positive or negative infinity for the case of vertical slopes. Therefore, we report, not the slope of the line, but instead the angle of the slope.”

In terms of a non-parametric check for other forms of modulation, our suggestion is to first consider what the candidate parametric models predict. For instance, bandwidth modulation is likely to produce curved functions in the relationship between the two conditions for each voxel. In such a case, the appropriate non-parametric check might be linearity (additive or multiplicative) versus non-linearity (bandwidth). More generally the non-parametric check is something to develop for each application in light of the specific set of candidate models. As quoted above, the section on the non-parametric model now says (Lines 242 - 249):

“The parametric model is crucial to the development of the non-parametric check and the specific parametric models under consideration should first be used to generate predictions for the check. The consequences of parametric assumptions are then considered in light of these predictions. In the case of a comparison between multiplicative gain versus additive shift, these two forms of tuning modulation can be distinguished using voxel-wise regression; if the modulation is multiplicative, a within-voxel plot comparing high- versus low-contrast should have a slope greater than 1, whereas if the tuning modulation is additive, this plot should reveal a slope equal to 1.”

Additional minor comments:

Lines 22-23: Fig. 1A shows slices through a 2D tuning profile (SF x orientation) – is this really analogous to changes in tuning as a function of cognitive state? Or is this just a more complete characterization of the (non)linear filter of the cell?

This is a good point, and we now clarify that the framework can be applied to either changes in cognitive state or changes in perceptual state (this is an example of changes in *perceptual* state). Indeed, the experimental manipulation we use in this study – stimulus contrast – is similarly better described as a manipulation of perceptual state than of cognitive state. We included this figure to show a more complete characterization of the cell’s filter, and to give an example of changes in the tuning for one stimulus feature across levels of some other feature, experimental manipulation, or perceptual state.

Lines 29-30: this introduces the idea of a ‘voxel tuning function’ without defining/describing how one might be measured

A clause has been added to describe the procedure (Lines 27 - 31):

“Nonetheless, fMRI studies reveal feature-selective tuning in voxels (e.g., in which a voxel’s BOLD response varies systematically as a function of a well-defined feature such as stimulus

orientation)^{5,6}, which is assumed to arise from a non-uniform distribution of tuning preferences across the neurons contributing to a voxel⁷.”

Fig. 1 caption: usually change in fluorescence is listed as uppercase $\Delta F/F$

The figure and caption have been adjusted accordingly.

Lines 98-99: low-contrast is assumed to be a ‘baseline’ condition – wouldn’t this be a lower-SNR scenario, so the model fits would be noisier? I’d think you’d want to use the highest-SNR data possible for model fitting. Perhaps the Bayesian modeling accounts for this somehow – regardless, the authors should add text convincing readers this isn’t something to be concerned about

Because the data are considered jointly, it does not matter which condition is labeled as the baseline. This is now explained (Lines 122 - 126):

“It is critical to note that the choice of baseline condition is arbitrary (e.g., the high-contrast condition could be labeled as the baseline) considering that the model is applied to both conditions conjointly. The only difference in terms of which condition is labeled as baseline is whether the modulation parameters are increases or decreases.”

Fig. 3: what ROI(s) is this? Number of subj? CI’s across...subj? voxels?

The caption has been updated with this information.

Line 131: reference to missing Figure C1

The reference has been updated to “Supplementary Figure 2”

Eq 4a: the parameter list prior to the = contains a parameter alpha, but the term to the right of = contains only a? The text in the preceding paragraph is also confusing about what the alpha and a parameters actually are – the authors should clarify.

There was a typo in that equation which has now been corrected.

Fig. 5A: is this plotting hierarchical model fits? Or data? Or both? This, I think, is the closest the authors get to plotting real data for evaluating their modeling procedure, but it’s very hard to understand what’s going on in the figure at present

This figure and its caption have now been revised substantially. Please see our earlier response.

Lines 265-266: “unreflective of the response properties of the underlying, individual neurons” – this doesn’t seem to be an accurate characterization of VTFs. Maybe changes in encoding properties measured with tools like fMRI don’t unambiguously demonstrate a particular coding mechanism, but I think it’s going a bit too far to say the tools give results that don’t reflect the underlying neural response properties at all, which is what’s implied at present

The sentence has been revised and now reads (Lines 366 - 368):

“Thus, voxel tuning functions have been interpreted as population-level properties of stimulus representations in the brain – population-level responses that cannot be used to infer tuning properties of the underlying, individual neurons^{10,12-14}.”

Lines 282-293: this paragraph of the discussion implies that SPM methods are more ‘fine-grained’ than multivoxel methods – I don’t understand the authors’ argument here; these are extremely different approaches that can’t easily be compared this way

In the course of editing, to provide space to expand on other points, we removed the referenced paragraph (which was not crucial to our arguments) from the Discussion.

Lines 306-307 (and this paragraph) attempt to distinguish the current approach from other encoding model methods, and point out that the choice of a baseline condition in encoding model techniques is arbitrary. However, isn’t that the case here too? What’s less arbitrary about picking a low-contrast baseline here as compared to the reports cited?

We apologize for our confusing use of the term baseline. As explained above, the INTM framework does not assume a baseline condition – it applied to both conditions at the same time. What we were referring to in this section is a standard practice with inverted encoding models in which the model is first applied to a baseline condition by assuming a specific neural tuning function with an arbitrarily chosen parameter values for width of tuning, prior to inverting the model. Regarding this issue, we now write (Lines 393 - 407):

“Encoding models can be applied to multiple conditions – for example through simulation³⁴, or through fitting to data and inverting³⁵ – thereby estimating how the channels are modulated by an experimental manipulation. Inverting an encoding model uses it as a decoder, estimating channel responses for other stimuli^{15,16,36} or estimating how channels are modulated in other conditions³⁷. But this use of inverted encoding models requires hard-wired (and arbitrary) assumptions about the shape of the channel responses in baseline conditions^{38,39}. These assumptions are even more restrictive than those made by the parametric model in INTM considering that they not only assume a specific class of shape (e.g., a circular Normal), but they require adopting a specific, user-selected parameter value for that shape (e.g., a particular value for the width of the circular Normal). These assumptions mean that the inferred channel modulation cannot be taken to reflect the modulations in underlying neural functions that would be measured with electrophysiology, as users of such models readily acknowledge^{14,38}. As a result, inverted encoding models have been used to pursue very different goals than the aim of the present study¹⁴ (i.e., understanding population- rather than neural-level representations).”

Lines 321-323: I don’t understand this sentence in context. Doesn’t the NMM framework involve an explicit form of the neural (and voxel) tuning functions?

We apologize for failing to make the distinction between class of shape (e.g., normal) versus parameter values of shape (e.g., width parameter). This section has been altered to read (Lines 417 – 421):

“Like inverted encoding models, the parametric analysis in INTM assumes a specific class of neural tuning shape (circular Normal). But unlike inverted encoding models, it avoids the need to assume particular parameter values for the shape (e.g., tuning width). The non-parametric slope analysis in INTM takes things a step farther by avoiding the need to assume a specific class of neural tuning shape.”

Line 369: I don't think Fig. 3 looking unimodal can be used in any way to support the use of a unimodal weight function for orientation preference within each voxel. It's likely there are a very large number of non-unimodal voxels that contribute. (for example, generating random mean responses for each voxel, then generating a synthetic data with noise; sorting voxels according to the max of a held-out dataset, and plotting the average response from the remaining data – the standard VTF approach applied to noisy synthetic data with no unimodal voxel tuning function – is likely to result in a unimodal VTF).

As discussed above, we now show individual voxel results, which are typically unimodal. However, this observed unimodality does not necessarily imply that the weight distribution is unimodal. It is for this reason that we developed the non-parametric check, which does not assume a unimodal weight distribution (see responses above). Our portrayal in Figure 2 (formerly Figure 3) was not intended to be evidence of unimodality but rather this was presented to promote conceptual understanding of how a voxel, with its many thousands of neurons, might end up preferring a particular orientation (i.e., the concept of a link between neural tuning and voxel tuning).

Line 443: what was/were the inter-trial intervals?

The following sentence has been added to the methods section (Lines 546 - 547):

“Inter-stimulus intervals ranged from 8000 – 12,000 ms in steps of 200 ms.”

pRF methods: what was the area of the screen mapped? What was/were the size(s) of the bars/wedges?

We have added to the Methods section (Lines 549 - 554):

“We mapped a circular area of the visual field, of radius 8° centered on a central fixation point. pRF mapping scans followed the protocol of Benson et al.⁵⁶. Briefly, natural images⁵⁷ were overlaid on pink noise and viewed through a series of central circular apertures (8° radius). Within one run per session, the apertures enabled view of either moving bars (2°) or rotating wedges (1/4 aperture) and rings that expanded and contracted (see stimulus software for details)”

ROIs: how were these defined? (I assume using pRF parameters) and which ROI(s) were used throughout? I think I saw one reference to primary visual cortex, but this should be made extremely clear

A new supplementary results section has been added to the Supplementary Information to clarify this point:

“Only voxels from V1 were analyzed, and only voxels whose population receptive fields overlapped with the stimulus (Methods). This resulted in 1,010 voxels, ranging from 90 to 188 across participants.”

Lines 545-546: why were the 3 pRF sessions analyzed independently? And how were the parameters combined across sessions? (or, instead, were the within-session parameters used to select voxels?)

The three sessions were analyzed separately because the voxels *might* be aligned differently with the neural tissue for each session. The power of the three sessions was exploited instead by analyzing each session separately and then using each session as an independent “test” of whether the voxel’s pRF was encompassed by the stimulus. The methods now clarify this (Lines 654 - 656):

“A voxel was retained only if a circle centered on its pRF with radius equal to two standard deviations was entirely contained by the grating stimulus in each of the three sessions.”

Lines 544-550: use of 2 SD-radius circle seems overly conservative. How many voxels were selected per participant/ROI?

We opted for a relatively conservative criterion in order to improve the responsiveness of voxels to the greatest extent possible while still retaining an adequate number of voxels. A new supplementary results section has been added to clarify how many voxels were retained:

“Only voxels from V1 were analyzed, and only voxels whose population receptive fields overlapped with the stimulus (Methods). This resulted in 1,010 voxels, ranging from 90 to 188 across participants.”

In the Results, the authors mention an “approximation to cross-validation” in the methods, but I couldn’t find any specific mention of this in the Methods.

The Methods now clarify which technique is the approximation with the following (Lines 685 - 688):

“All model comparison was done with Pareto-smooth importance sampling, “leave-one-out cross-validation plus”, which is a technique for approximating cross-validation²⁸. We refer readers to the original report for details of this technique.”

Reviewer #2 (Remarks to the Author):

1. This review is tricky because I think the basic aim of this paper (forward modelling of fMRI data) is really important, and the general approach taken to this problem is excellent and sophisticated, but the example application to the simple case of distinguishing multiplicative vs additive models of visual contrast is a somewhat trivial, and the proof for the “non-parametric check” in the supplementary material undermines the rationale for the parametric model (in showing that multiplicative a versus additive modulation can be distinguished simply by slope of activation plots, without fitting data). Is there another empirical test to which the parametric model could be applied, to demonstrate the need for parametric model fitting, i.e. that cannot be shown a priori to be testable with a simpler data analysis technique? Or if I am missing the point, perhaps the authors could explain why the parametric model is so important for the empirical case they chose?

Thank you for pointing out these issues. In our initial submission we failed to explain that the INTM framework (which was called NMM in our initial submission) can be applied to more than just additive versus multiplicative modulation and we did not make clear how even this additive/multiplicative comparison is extremely challenging considering that the data are clearly additive when analyzed in terms of average responses (the observed average voxel tuning curves in Figure 4A appear to be perfectly additive). We also failed to explain the relationship between the parametric and non-parametric models and how the latter relies heavily upon the former.

We now make clear several places that we did in fact consider other forms of tuning modulation. Although the analyses we present focused on a comparison between an additive shift and multiplicative gain, we also considered changes in tuning bandwidth and orientation preference shifts. In the present dataset, those forms could be rejected even without model comparison due to their inability to capture gross trends in the data (both these forms predict no average change in the BOLD signal). We now make this clear in the presentation of the parametric model writing (Lines 138 - 146):

“Within the INTM framework, each form of tuning modulation is instantiated by incorporating additional parameters that modulate one or more parameters in the baseline voxel tuning function (Equation 3, Figure 1C-D). In this validation test case as applied to visual contrast manipulations, we considered two forms of tuning modulation that are capable of producing an average increase in neural activity: multiplicative gain and additive shift. Other forms of neural tuning modulation, such as bandwidth changes and tuning preference shifts, could be implemented within the INTM framework, but these forms were not included in the candidate set of models for this test case because they do not change average neural activity (i.e., they are not viable models for the neural modulation that underlies changes in visual contrast).”

Later in this section of the paper, we highlight how an end-user of the framework could address these other kinds of modulation (Lines 168 – 173):

“For other applications, shifts in orientation preferences could be modeled by allowing φ_v to change across conditions, and a change in concentration could be modeled by allowing κ_v to change across conditions. In the current dataset, these forms were not pursued because neither allows contrast to alter the average activity of a voxel across orientations – an effect that was clearly present in the empirical data (Figure 4A) – and so these forms could be rejected without explicit modeling.”

In the section of the manuscript that develops the parametric model, we now explain how this is a particular challenging test case, writing (Lines 147 - 158):

“In this test case, multiplicative gain is known to be “ground truth”—that is, changes in visual contrast induce multiplicative gain in single-neuron tuning functions^{19–21}. However, in seeming contradiction to multiplicative modulation, an examination of raw voxel tuning functions suggests an additive increase with increases in visual contrast (Figure 4A,B). Nevertheless, averages can be misleading, and a formal model comparison between additive shift and multiplicative gain is required that considers each voxel separately, particularly when considering that many of the voxels contributing to the average exhibit poor tuning. For

completely untuned voxels, additive shift and multiplicative gain are indistinguishable and so the average will necessarily look additive. If INTM works, it should leverage differences between voxels (i.e., capitalize on the small proportion of well-tuned voxels), to reach the conclusion that multiplicative gain is the more likely form of tuning modulation despite what the average BOLD signal appears to show."

In terms of a non-parametric check for other forms of modulation, our suggestion is to first consider what the candidate parametric models predict. For instance, bandwidth modulation is likely to produce curved functions in the relationship between the two conditions for each voxel. In such a case, the appropriate non-parametric check might be linearity (additive or multiplicative) versus non-linearity (bandwidth).

More generally the non-parametric check is something to develop for each application in light of the specific set of candidate models. The section on the non-parametric model now says (Lines 242 - 249):

"The parametric model is crucial to the development of the non-parametric check and the specific parametric models under consideration should first be used to generate predictions for the check. The consequences of parametric assumptions are then considered in light of these predictions. In the case of a comparison between multiplicative gain versus additive shift, these two forms of tuning modulation can be distinguished using voxel-wise regression; if the modulation is multiplicative, a within-voxel plot comparing high- versus low-contrast should have a slope greater than 1, whereas if the tuning modulation is additive, this plot should reveal a slope equal to 1."

Finally, we note that the non-parametric check is likely to be less statistically powerful, similar to other non-parametric tests, in contrast to parametric tests. The non-parametric test is looking for qualitative data trends (e.g., slope of 1 versus greater than 1) whereas the parametric model characterizes each voxel separately, with different parameters for each voxel. For instance, in the non-parametric check, a completely untuned voxel will necessarily have a slope of 1 even if the underlying truth is multiplicative modulation. The non-parametric check lumps together all slopes, regardless of how well-tuned the voxel is, asking whether the distribution of slopes is generally greater than 1. In contrast, the parametric model assesses the tuned-ness of each voxel separately, and in doing so the model comparison naturally discounts as uninformative any untuned voxels. Put another way, for untuned voxels, the additive/multiplicative parametric models properly make the same prediction (slope of 1) whereas the non-parametric check is asking whether the slope differs even for these untuned voxels. This will handicap the non-parametric check in terms of statistical power.

2. I could not find code for the models on the original OSF link given, but thank the authors (and editor) for providing the link in a separate email – I hope this link will be included in the final version of the paper. Given that my review was already overdue because of the delay entailed by me asking for the code, I did not have time to run stan and the associated functions, but I did look through them, and it helped better understand the models. If there is a revised submission (and more time to update the code repository as the email suggested), I might have time to test the code then. One minor observation: I think there may be a literal-code-and-paste error in the README.md of <https://gitlab.com/psadil/>, where the code below the section for the “Additive” model looks identical to that for the “Multiplicative” model above?

The paper includes a “Code Availability” section at the end of the Methods (Line 752), which states that:

Software to perform the analyses is available on the Open Science Framework (<https://osf.io/93ynm/;10.17605/OSF.IO/93YNM>).

There was a typo in the README.md, which has been fixed.

3. Line 95: what if one allows neural tuning functions to vary in shape within a voxel? This seems possible in reality – so if this assumption is dropped, can any inferences still be made, or does the whole framework become inestimable (at least without strong priors on the variation in shapes within a voxel)?

This is an excellent question. In response, we went back to our proof regarding the non-parametric slope test and realized that the slope test did not need to assume that all neurons within a voxel have the same shape. Instead, every neuron can differ from its neighbors in arbitrary ways, and even take on multimodal shapes. This section of the supporting information document has been completely rewritten to reflect this, with a new proof regarding the generality of the non-parametric slope test (it’s even more non-parametric now). We do not put the new proof in this cover letter, but the final paragraph of the proof now reads:

"This proof makes no assumptions about the shapes of the neural tuning functions and it makes no assumptions that the neurons contributing to each voxel have the same shape. Each neuron is allowed to have its own unique tuning function. The one key assumption made in the slope test (an assumption that is shared with the parametric model), is that all neurons contributing to a particular voxel have the same multiplicative modulation or the same additive modulation. If the modulation constant varies across neurons contributing to a voxel, then it is possible (but perhaps unlikely) that multiplicative modulation could produce a slope of 1.0 and that additive modulation could produce a slope greater than 1.0. For instance, if the additive modulation was greater for the neurons that preferred stimuli that were more-preferred by the voxel, additive modulation could produce a slope greater than 1.0. Analogously, if the multiplicative modulation was smaller for the neurons that preferred stimuli that were more-preferred by the voxel, multiplicative modulation could produce a slope of 1.0. Such confounding relationships between neural preferences and voxel preferences may occur by chance for some voxels, but there is no obvious reason to expect such a systematically confounding relationship to occur for most voxels. Thus, it is likely that even this assumption could be relaxed, provided that

heterogeneity of modulation magnitude is uniformly applied across the neurons contributing to a voxel."

Regarding this issue, the main text now reads (Lines 181 - 192):

"The parametric model assumes that all neurons within a voxel share a common shape for their tuning function (homogeneity of tuning shape), which is unlikely to hold strictly true. Nevertheless, some degree of tuning shape heterogeneity should not invalidate inferences drawn about the form of tuning modulation, provided that tuning function shape does not differ systematically as a function of the orientation preference of the neurons. That is, there could exist a range of tuning function shapes, but if that heterogeneity occurs to the same extent for neurons centered on all preferred orientations, then it should not lead to scenarios where one form of modulation mimics another at the voxel level. Nevertheless, to assess whether such heterogeneity posed a problem when adjudicating between the forms of tuning modulation, we also developed a non-parametric check that does not make this assumption. If the non-parametric check supports the same qualitative conclusion as the parametric model, this suggests that homogeneity of tuning shape assumption was adequate."

4. In the paragraph beginning on line 134, is it possible that the K_v parameter does change with contrast, and it *could* change the average activity of a voxel, if one dropped the unit-normalisation of the integral of the von Mises function(f)? It seems likely to me that some neurons with a preference close to the stimulus orientation could change their concentration as contrast changes, while other neurons with preferences further away do not change at all?

Without needing to drop the unit-normalisation assumption, an effect like this could be achieved by allowing combinations of parameters to change – for example, allowing a change in concentration along with either an additive shift or multiplicative gain change. The penultimate paragraph of the discussion now addresses this limitation, and reads (Lines 488 - 501):

"The second key limitation is that INTM can differentiate only between models that are formally included in the model comparison process. If the "true" form of tuning modulation is not included in the set of possibilities, the results may mislead researchers into conflating the winning form of modulation with the true form. For instance, in the present study, the technique contrasted tuning modulation models in which just one form of modulation occurred but did not consider more complex situations in which multiple forms of tuning modulation could have occurred. For example, it is known from electrophysiology that with extended adaptation, orientation tuning functions can both widen and shift their preferred orientation⁵³. Since that more complex model, in which tuning widened and shifted within every trial, was not included in model comparison, INTM is silent about whether that effect occurred. Note that such combinations of tuning modulation could be included in the model comparison process, but when attempting to adjudicate between complex models it would be critical to use model recovery simulations, specifically data-informed recovery, to determine if the collected data were sufficiently constraining."

5. On line 384, it is stated "...applying NMM to explore neuromodulations beyond multiplicative gain and additive shift, further model recovery simulations would be required." It would really strengthen the paper (though I suspect the authors will groan, and argue this is future work!) if they explored model recovery for a much larger range of variables in the models (eg Point 4 above). I suspect that the spatial limitations of fMRI will mean that the "inverse" problem of inferring neural properties from voxel-level data becomes too ill-posed, such that useful inferences are only possible for a very small subset of possible hypothetical neural effects (like multiplicative vs additive effects). Perhaps I am wrong, but I think this would be a very influential paper: i.e, documenting what type of neural hypotheses can be distinguished by fMRI and what types can NOT (once the complexity of the mapping is modelled, as the authors do here). This would back-up the authors' claim on line 402 that "Despite the simplicity of the test case, the method should be applicable to a broad range of domains and manipulations."

In this report we have sought to provide a framework within which end users can compare a range of theoretically interesting alternative forms of neural tuning modulation. However, every application of the framework will be different and is likely to require model recovery to ascertain whether the empirical data are sufficiently constraining. Regarding this point, the third to last paragraph of the discussion now reads (Lines 469 - 487):

"Despite its generality, INTM has two limitations that must be plainly acknowledged. The first is that INTM requires that the models under consideration make distinct predictions at the level of the voxel. For instance, the parametric and non-parametric INTM techniques both assume that the magnitude of tuning modulation is the same for all neurons that contribute to the voxel response. These techniques may be robust to relaxing this assumption if modulation magnitude difference apply uniformly across the weight distributions that maps neural preferences into voxel preferences. But if the magnitude of modulation varies systematically, there might no longer be distinct predictions at the level of the voxel. For example, if the magnitude of the additive shift is larger for neurons that prefer the same orientation as the voxel, this unlikely circumstance could produce a voxel tuning function that is indistinguishable from a constant magnitude of multiplicative gain. The inability to discriminate these two scenarios exemplifies the fundamental inverse problem that INTM cannot solve: INTM cannot determine the parameters of tuning functions for individual neurons (indeed, it is not designed to). Instead, INTM operates with distributions of neurons (grouped by, e.g., region-of-interest, voxel, feature preferences), and so it can only compare models whose distributions of tuning functions change in ways that are distinct. With the example of a magnitude of additive shift that happened to covary with the weight distribution, this might occur for some voxels, but is unlikely to occur with most voxels, demonstrating the need to use all of the voxels in the model comparison."

A complete list of possible tuning modulations is large (e.g., besides changes in additive gain, multiplicative gain, orientation preference, or tuning width, there are an additional six models that include two of these changes, three models with all three forms of change, and another with all four). Whether it is possible to differentiate between all models will depend on the strength of tuning, the quality and quantity of the data, and the accuracy of assumptions. We have explicitly avoided making concrete claims regarding which forms of tuning modulation will be identifiable. This is because identifiability depends on more than the relationship between neural and BOLD changes; it depends

on factors that will be particular to each experiment – like the manipulation of interest, stimulus feature, study population, etc. In our discussion, we have provided the following justification for keeping the current manuscript focused (Lines 444 - 465):

“The techniques developed here were tailored to uncover changes in orientation tuning caused by stimulus contrast, relying on assumptions that will not be appropriate in all studies. We highlight these assumptions here and use them to clarify the distinction between the framework more generally versus specific applications of the framework. First, the models assumed that voxel tuning functions are a linear combination of the activity of the underlying neural tuning functions. This assumption is reasonable in some experiments^{42,43}, but the relationship between neural firing rate and BOLD signal is not perfectly linear^{47,48}. Second, the techniques implicitly assumed, through the general linear model used to estimate voxel activity, a hemodynamic response function that is shared across all voxels in all participants. Although canonical, that assumption is erroneous e.g., 49+. Next, the parametric analysis approximated the distribution of neurons tuned to orientations within each voxel with a circular Normal distribution, which is unimodal and periodic. This approximation provided a convenient formula for deriving the voxel tuning function, and the resulting function matched the data (e.g., individual voxel tuning functions in Figure 4B appear unimodal). However, this assumption will not hold for other types of stimuli, other brain regions, or for other scan parameters (e.g., an aperiodic stimulus like pitch would require an aperiodic tuning function).

Violations of assumptions may bias model comparison in some cases, but regardless none of the assumptions are intrinsic to the framework, and there are several ways of relaxing each of them. For example, more complex weight distributions such as multimodal functions could be used in the parametric model. Alternatively, we have shown that assumptions about the weight distributions can be bypassed by a technique like the non-parametric slope analysis.”

Minor points

6. In the Discussion, it is stated that “Like biophysical models, NMM connects the latent neural activity to the observed BOLD, by assuming a linear coupling that is justified in our experimental design...” In most current biophysical models, the BOLD response to neural activity has several nonlinearities, even if a linear convolution model is often assumed in fMRI analysis (so I don’t think the “Like biophysical models” bit is correct). This linear approximation may be sufficient for the temporal profile of responses to successive brief stimulations (used in most fMRI designs), at least if the time between brief neural bursts (as determined by SOA) is long enough, but the BOLD response has been claimed to be a nonlinear function of other variables like the duration of neural activity and possibly stimulus contrast (see Vazquez & Noll, 1998, NeuroImage). Even putting to one side debates about whether nonlinearities as a function of contrast are neural or haemodynamic in origin, I’m not sure many other types of nonlinearity matter for the present model, which is a spatial rather than temporal model – i.e., has no concept of duration of neural activity, so could not be applied to designs with different stimulus durations (without extension to the temporal domain). In short, while it’s good to acknowledge that the current model assumes a linear neural-to-BOLD mapping, it might help for the reader to consider precisely what types of nonlinearities (i.e, as function of what types of experimental variables) might be relevant to the model’s predictions.

These are good points for discussion. It was not our intention to suggest that biophysical models rely on assumptions of linearity. Instead, we meant to suggest that biophysical models provide a specific (and testable) mapping between neural activity and the BOLD signal, and that such an explicit link is also true in our framework. Indeed, the form of this coupling is nonlinear in most biophysical models. You are correct to point out that the assumption of linearity, such as with using a GLM as applied to the HRF, is justified by our experimental design, rather than being intrinsic to the INTM framework. We have adjusted this line as follows (Lines 412 – 415):

“Like biophysical models, INTM connects the latent neural activity to the observed BOLD. It does so by assuming a linear coupling that is justified given our experimental design and the goals of our modeling procedure (we discuss potential violations of this assumption in more detail below)^{42–45}.”

The paragraph in the discussion concerning violations of assumptions continues with (Lines 465 - 468):

As an another example, although nonlinear relationships between the neural firing rate and the BOLD signal have been documented, multiple mathematical accounts have been proposed to explain these nonlinearities^{48,50,51}, and these relationships could be incorporated into INTM.”

7. Line 193: isn’t “orthogonal regression” the same as what is commonly known as “Deming regression”? This doesn’t necessarily need to be changed; just might be better known to some by this name (cf von Mises distributions!).

Deming regression is indeed closely related to orthogonal regression. It is our impression that some authors reserve the label “Deming” regression to situations in which the ratio between the variances

of the two sets of observations is different than 1 but known or assumed (e.g., that the trial-wise noise in the high-contrast condition is 1.1x the trial-wise noise in the low-contrast condition), and orthogonal regression for a situation in which the ratio is assumed to be 1. In either case, we have elected to retain the label “orthogonal” regression as this is more descriptive of the slope analysis, which implicitly assumed a ratio of 1 (i.e., gave equal weight to errors in both the x and y direction).

That said, we note that the supplementary materials include a Bayesian model that estimates both variances directly. In describing this model in the supplementary material, we refer to it as neither the “Deming” nor “Orthogonal” regression.

8. Line 234, for those less familiar with Bayesian model comparison, could some intuition be given for the “expected log pointwise predictive density”, eg guidelines for how impressive 14.5 standard errors is?

A line has now been included to relate the model comparison result to a frequentist approach (Lines 330 - 332):

“Adopting a standard significance threshold (e.g., $\alpha = 0.05$ or 1.65 standard errors) would suggest that this difference in predictive ability is poorly accounted for under the null hypothesis that the two forms of tuning modulation are equally predictive.”

Reviewer #3 (Remarks to the Author):

The authors have proposed an interesting and novel model to link a voxel’s BOLD response to neural activity. As they have correctly mentioned, their model can have applications in investigating the effect of cognitive states on perception. However, their model is based on some simplifying assumptions, and it is not clear how they will impact the results. One of these assumptions is the homogeneity of the changes of the neural responses, which is far from reality. Even in the references they have provided for their “ground truth” case, the observations have suggested that not all the neurons show a boost in orientation tuning function with contrast, and there are neurons that reach saturation at very low contrasts, below the 50% contrast level used in this study; the point that has not been even mentioned after citing these studies. They have admitted to this limitation in their discussion but have neither discussed the possible misjudgments that may arise due to this simplifying assumption nor the way the non-linearity of the neural responses can be incorporated in the model. They argue that even with this simplifying assumption the NMM has been able to detect the correct model, but picking the right model from two options and in a simple case cannot be a strong proof for the validity of the model in more complicated situations and especially when it is supposed to be used for discovering neural dynamics in novel conditions. Moreover, this limitation has not been touched upon in the abstract. The given image of the NMM in the abstract as a complete model that can be used in all conditions and without limitations is somehow misleading.

Thank you for these comments. In this revision we have made many changes to make it clear what the model assumes, noting that different assumptions are made for different models (e.g., the parametric model makes more assumptions than the non-parametric slope test). The name of the model is changed slightly, and regarding these issues, the abstract now says (Liens 4 – 10):

“...we developed an analysis framework called Inferring Neural Tuning Modulation (INTM) for “peering inside” voxels. Precise specification of neural tuning from the BOLD signal is not possible. Instead, INTM compares theoretical alternatives for the form of neural tuning modulation that might underlie changes in BOLD across experimental conditions. The most likely form is identified via formal model comparison, with assumed parametric Normal tuning functions, followed by a non-parametric check of conclusions.”

The end of the section describing the parametric model now says (Lines 181 - 192):

“The parametric model assumes that all neurons within a voxel share a common shape for their tuning function (homogeneity of tuning shape), which is unlikely to hold strictly true. Nevertheless, some degree of tuning shape heterogeneity should not invalidate inferences drawn about the form of tuning modulation, provided that tuning function shape does not differ systematically as a function of the orientation preference of the neurons. That is, there could exist a range of tuning function shapes, but if that heterogeneity occurs to the same extent for neurons centered on all preferred orientations, then it should not lead to scenarios where one form of modulation mimics another at the voxel level. Nevertheless, to assess whether such heterogeneity posed a problem when adjudicating between the forms of tuning modulation, we also developed a non-parametric check that does not make this assumption. If the non-parametric check supports the same qualitative conclusion as the parametric model, this suggests that homogeneity of tuning shape assumption was adequate.”

The start of the section describing the non-parametric slope test now says (Lines 234 – 255):

“The parametric model assumed normal functions for the NTFs and weight distributions and additionally assumed homogeneity of NTF shape within a voxel. These assumptions will often be violated, but such violations might not pose a problem for model comparison, particularly if the violations of these assumptions are uniformly applied across the stimulus dimension of interest. Nevertheless, as a complement to parametric model comparison, we developed a more qualitative non-parametric check of the empirical data. If this check supports the same conclusion as the parametric model, this suggests that any violations of parametric assumptions were insufficient to alter the theoretical conclusions.

The parametric model is crucial to the development of the non-parametric check and the specific parametric models under consideration should first be used to generate predictions for the check. The consequences of parametric assumptions are then considered in light of these predictions. In the case of a comparison between multiplicative gain versus additive shift, these two forms of tuning modulation can be distinguished using voxel-wise regression; if the modulation is multiplicative, a within-voxel plot comparing high- versus low-contrast should have a slope greater than 1, whereas if the tuning modulation is additive, this plot should reveal a slope equal to 1. Furthermore, these predictions do not rely on the parametric assumptions. For instance, if a voxel is multimodal, preferring more than one orientation, this would only serve to rearrange the order of the orientations in the regression plot, but the qualitative distinction between a slope of 1 versus a slope greater than 1 would still map onto additive shift versus multiplicative gain. The generality of the slope test across all shapes of

tuning function and all shapes of weight distribution is mathematically proved in Supplementary Methods Equations 1-4.”

In response to your comments, the proof in Equations 1-4 of the supplementary methods is new. This modified proof now allows that every neuron can take on a unique tuning shape and differ from its neighbors within the same voxel (including the possibility of completely untuned neurons). However, both the parametric and non-parametric models must assume that the magnitude of modulation for a given voxel is the same for all neurons within the voxel. As you correctly point out, some neurons may saturate in their response more easily than others, and thus the magnitude of modulation is likely to be different for different neurons. This assumption is addressed in the discussion, which says (Lines 469 – 487):

“Despite its generality, INTM has two limitations that must be plainly acknowledged. The first is that INTM requires that the models under consideration make distinct predictions at the level of the voxel. For instance, the parametric and non-parametric INTM techniques both assume that the magnitude of tuning modulation is the same for all neurons that contribute to the voxel response. These techniques may be robust to relaxing this assumption if modulation magnitude difference apply uniformly across the weight distributions that maps neural preferences into voxel preferences. But if the magnitude of modulation varies systematically, there might no longer be distinct predictions at the level of the voxel. For example, if the magnitude of the additive shift is larger for neurons that prefer the same orientation as the voxel, this unlikely circumstance could produce a voxel tuning function that is indistinguishable from a constant magnitude of multiplicative gain. The inability to discriminate these two scenarios exemplifies the fundamental inverse problem that INTM cannot solve: INTM cannot determine the parameters of tuning functions for individual neurons (indeed, it is not designed to). Instead, INTM operates with distributions of neurons (grouped by, e.g., region-of-interest, voxel, feature preferences), and so it can only compare models whose distributions of tuning functions change in ways that are distinct. With the example of a magnitude of additive shift that happened to covary with the weight distribution, this might occur for some voxels, but is unlikely to occur with most voxels, demonstrating the need to use all of the voxels in the model comparison.”

The Supplementary Methods goes into greater depth on this issue and now says:

"This proof makes no assumptions about the shapes of the neural tuning functions and it makes no assumptions that the neurons contributing to each voxel have the same shape. Each neuron is allowed to have its own unique tuning function. The one key assumption made in the slope test (an assumption that is shared with the parametric model), is that all neurons contributing to a particular voxel have the same multiplicative modulation or the same additive modulation. If the modulation constant varies across neurons contributing to a voxel, then it is possible (but perhaps unlikely) that multiplicative modulation could produce a slope of 1.0 and that additive modulation could produce a slope greater than 1.0. For instance, if the additive modulation was greater for the neurons that preferred stimuli that were more-preferred by the voxel, additive modulation could produce a slope greater than 1.0. Analogously, if the multiplicative modulation was smaller for the neurons that preferred stimuli that were more-preferred by the voxel, multiplicative modulation could produce a slope of 1.0.

Such confounding relationships between neural preferences and voxel preferences may occur by chance for some voxels, but there is no obvious reason to expect such a systematically confounding relationship to occur for most voxels. Thus, it is likely that even this assumption could be relaxed, provided that heterogeneity of modulation magnitude is uniformly applied across the neurons contributing to a voxel."

The other point you make is that a comparison between just two models is relatively. We agree, and in this revision we now highlight that the framework can be used to compare many different models. Furthermore, we now highlight that we considered more models than just two for this validation case but that other models were immediately rejected for failing to explain changes in the mean BOLD response with changes in visual contrast. This is first mentioned in the presentation of the parametric model, which says (Lines 168 – 173):

"For other applications, shifts in orientation preferences could be modeled by allowing ϕ_v to change across conditions, and a change in concentration could be modeled by allowing κ_v to change across conditions. In the current dataset, these forms were not pursued because neither allows contrast to alter the average activity of a voxel across orientations – an effect that was clearly present in the empirical data (Figure 4A) – and so these forms could be rejected without explicit modeling."

This issue is again considered in the discussion, which says (Lines 488 – 501):

"The second key limitation is that INTM can differentiate only between models that are formally included in the model comparison process. If the "true" form of tuning modulation is not included in the set of possibilities, the results may mislead researchers into conflating the winning form of modulation with the true form. For instance, in the present study, the technique contrasted tuning modulation models in which just one form of modulation occurred but did not consider more complex situations in which multiple forms of tuning modulation could have occurred. For example, it is known from electrophysiology that with extended adaptation, orientation tuning functions can both widen and shift their preferred orientation⁵³. Since that more complex model, in which tuning widened and shifted within every trial, was not included in model comparison, INTM is silent about whether that effect occurred. Note that such combinations of tuning modulation could be included in the model comparison process, but when attempting to adjudicate between complex models it would be critical to use model recovery simulations, specifically data-informed recovery, to determine if the collected data were sufficiently constraining."

There is also some information about the fMRI analysis that seems to be missing from the manuscript. It has not been stated clearly from which area the voxels have been selected. Is figure 3 representative of voxels of the V1 area or V1-V3? And how many voxels were left after applying a threshold in figure 5?

Thank you for pointing out this oversight. In the main manuscript, we have expanded our discussion of population receptive field mapping to specify that we focused on V1. It now reads (Lines 650 – 656):

“The resulting pRF parameters determined whether a voxel would be retained for analyses. The pRF resembles an isotropic, bivariate Gaussian. The three pRF sessions were analyzed separately, resulting in three sets of three pRF parameters per voxel. Within a set of parameters, two indicate the center location of the pRF, and the third determines its size – the standard deviation of the Gaussian. A voxel was retained only if a circle centered on its pRF with radius equal to two standard deviations was entirely contained by the grating stimulus in each of the three sessions.”

We have additionally included a brief subsection of the Supplementary Results to describe how many voxels remain after thresholding:

“Only voxels from V1 were analyzed, and only voxels whose population receptive fields overlapped with the stimulus (Methods). This resulted in 1,010 voxels, ranging from 90 to 188 across participants.”

I also suspect that the additive mechanism seen in figure 3 arises due to including all the voxels and excluding the less active voxels with the same thresholding approach may change this figure. Have the authors tried regenerating this figure with only the most active voxels?

We agree with this suspicion about Figure 3 and the requested analysis is essentially captured by the orthogonal regression results in Figure 5, which shows that the voxels that are more responsive (high beta values) are also the voxels that reveal more a more clearly multiplicative effect (slope greater than 1). However, justifying the exclusion of voxels that are “less” active can be difficult in practice, considering that there are many ways to define activity that may be equally valid (it becomes a problem of user-flexibility and could invite choosing a threshold that gives rise to the desired results). When developing this framework, one of our goals was to retain as many voxels as would be feasible. We discuss this in a new Results paragraph (Lines 300 - 312):

“Voxel tuning functions were first analyzed with a standard approach of “averaging” across aligned voxels (Figure 4)¹⁰, which is an appropriate technique if all voxels have similar tuning functions. This “binning” alignment technique appears to reveal an additive shift with contrast, in contrast to what is known from electrophysiology. This additive shift is caused by the inclusion of many voxels with poor orientation tuning, i.e., with an almost uniform distribution of neural tuning functions across preferred stimulus values. When the BOLD response is aggregated across a many such “undiagnostic” voxels, the true form of tuning modulation is obscured (i.e., it appears additive rather than multiplicative). One approach to mitigating this variability would be to adopt experiment-specific thresholds, excluding some subset of “relatively” untuned or unresponsive voxels. However, it would be difficult to justify such thresholding for each experiment, and more importantly excluding voxels throws away information (e.g., about the average noise in voxels’ responses). The two techniques of INTM avoid thresholding, even while using the well-tuned “diagnostic” voxels to identify the most likely form of tuning modulation.”

Minors:

- The purpose of the proposed tool is to link voxel's response to neural response and it seems to me that the NMM name that only focuses on neuromodulation does not communicate this purpose.

We agree that the name of the framework and we have changed it to "Inferring Neural Tuning Modulation" (INTM). Please see earlier responses to reviewers for an expanded discussion of this. Regardless of the name, we now make it clear that the goal of technique is not to reveal the neural response precisely, but only to characterize the likely form of neural tuning modulation. Although we link the neural response on the voxel response in formulating the parametric model, the goal is to use this link to adjudicate between different forms of changes to neural tuning. To clarify this point, we have revised the following paragraph in the discussion (Liens 385 – 411):

"Several techniques have previously been developed to peer inside the voxel, but none of them statistically compares alternative theories of tuning modulation using a model that explicitly links neural tuning with voxel tuning. For example, biophysical models specify how the BOLD signal arises from the often nonlinear coupling between neural activity and vasculature^{32,33}. But, despite this biological realism, these models have not yet been used to examine changes in neural tuning functions. In contrast to biophysical models, encoding models of fMRI data delve inside the voxel by modeling how components comprising a voxel (e.g., sub-voxel "channels") transform stimuli into voxels' activity^{5,15–17}, and, promisingly, enable researchers to use the BOLD signal to uncover features of neural tuning¹⁸. Encoding models can be applied to multiple conditions – for example through simulation³⁴, or through fitting to data and inverting³⁵ – thereby estimating how the channels are modulated by an experimental manipulation. Inverting an encoding model uses it as a decoder, estimating channel responses for other stimuli^{15,16,36} or estimating how channels are modulated in other conditions³⁷. But this use of inverted encoding models requires hard-wired (and arbitrary) assumptions about the shape of the channel responses in baseline conditions^{38,39}. These assumptions are even more restrictive than those made by the parametric model in INTM considering that they not only assume a specific class of shape (e.g., a circular Normal), but they require adopting a specific, user-selected parameter value for that shape (e.g., a particular value for the width of the circular Normal). These assumptions mean that the inferred channel modulation cannot be taken to reflect the modulations in underlying neural functions that would be measured with electrophysiology, as users of such models readily acknowledge^{14,38}. As a result, inverted encoding models have been used to pursue very different goals than the aim of the present study¹⁴ (i.e., understanding population- rather than neural-level representations). Beyond inverted encoding models, other researchers have compared the qualitative predictions of different forms of tuning modulation^{34,40,41}, analogous to the logic underlying INTM's non-parametric check. However, these techniques neither fit the alternative encoding models to empirical data, nor account for model flexibility."

- Von Mises has been used before for model fitting on the orientation tuning curves. I suggest citation of Swindale 1998.

We thank the reviewer for pointing out this reference. It has been included (Reference 23).

- Title of figure 2 does not convey the reason behind the analysis. The analysis has been done to demonstrate that von Mises weight function is a reasonable model, but the title suggests that it is only a simulation of orientation preferences and the text does not clarify the purpose either. It did not become clear to me until I reached the methods section.

The Figure's title has been updated to clarify its point.

"Orientation preferences of simulated voxels can be approximated with unimodal distributions"

- Line 131, Figure C1 does not exist.

The reference has been updated to "Supplementary Figure 2"

- Line 147, how the "weakly informative" priors have been selected? Have the parameters been set based on the literature?

The scale of allowed values was based on the literature very loosely (e.g., we did not allow gain to be a 1000-fold effect), but we used a range that was larger than would be truly plausible. We have clarified our use of "weakly informative" as follows (Lines 204 - 207):

"First, synthetic data were generated with each model using "weakly informative" priors (Supplementary Methods). These priors were not designed to reflect precise experimental knowledge but instead simply provide a rough scale for each parameter individually."

- Line 171, the shape of the neural tuning function and the weight distribution do not seem to be totally irrelevant. According to the given proof in the supplementary methods, $q(r)$ function, which is the convolution of these two, should be an invertible function for the non-parametric check to be true.

The proof has been changed to be much more general and no longer requires that the function be invertible. This was achieved by replacing the convolution with a sum over the discrete number of neurons contributing to the voxel (see Equations 1-4 of the Supplementary Methods and the associated text).

- Figure 5A: Contrary to the caption content, some slopes seems to be larger than 1 in the additive model figure. Adding a histogram of the slopes for the additive model as well may help resolve this confusion.

Please note that the subfigures in Figure 5 have been rearranged and so the reviewer's comments refer to what are now the second and third panels of Figure 5B.

The appearance of slopes greater than 1 in the additive case may have been a visual illusion; up to numerical precision, the slopes are all 1. Our previous figure likely obscured this by plotting the lines with different lengths. Now, each voxel's line is the same length across all panels, and we hope that this clarifies the comparison.

- Supplementary material, equation 6, wrong sign: $-\alpha$

We thank the reviewer for pointing this out.

Reference: Swindale, Nicholas V. "Orientation tuning curves: empirical description and estimation of parameters." *Biological cybernetics* 78.1 (1998): 45-56.

REVIEWERS' COMMENTS:

Reviewer #2 (Remarks to the Author):

I think the authors have done a good job of addressing my and the other reviewers' comments. I only have some very minor remaining ones:

On the OSF link, the model framework is still called "neuromodulation modelling", despite the revised name in the manuscript. I appreciate that the authors are creating an R package and it could be a pain to rename, so I am not insisting on this, but just point out in case they had forgotten (eg on the OSF site, they could at least mention the two names)

line 287 "produce results" = "produces results"

line 330 - formatting error on "density28. a significance"

For the record, I still think it unlikely that the magnitude of tuning modulation is the same for all neurons that contribute to the voxel response. I accept that the specific example they raise is unlikely, ie that the "magnitude of the additive shift is larger for neurons that prefer the same orientation as the voxel", but it would be reassuring to confirm through simulation that their approach still works if modulation magnitude differences apply uniformly (or randomly) across the weight distribution. But I'm not going to insist on this either!

Congratulations to the authors on great work.

Reviewer #3 (Remarks to the Author):

The revised manuscript has significantly improved in comparison to the older version, in my opinion. The authors have clearly articulated the limitations of their model, and the additional figures and analysis have strengthened their argument. In general, my concerns have been addressed, and I have no further comments.